# Bioinspired design of Na-ion conduction channels in covalent organic frameworks for quasi-solid-state sodium batteries

Yingchun Yan [1], Zheng Liu [1] ✉, Ting Wan[1], Weining Li[1], Zhipeng Qiu [1], Chunlei Chi[1], Chao Huangfu[1], Guanwen Wang[1], Bin Qi[1], Youguo Yan [1] ✉, Tong Wei[1] & Zhuangjun Fan [1] ✉

Solid polymer electrolytes are considered among the most promising candidates for developing practical solid-state sodium batteries. However, moderate ionic conductivity and narrow electrochemical windows hinder their further application. Herein, inspired by the $Na^+/K^+$ conduction in biological membranes, we report a ($-COO^-$)-modified covalent organic framework (COF) as a Na-ion quasi-solid-state electrolyte with sub-nanometre-sized $Na^+$ transport zones (6.7–11.6 Å) created by adjacent $-COO^-$ groups and COF inwalls. The quasi-solid-state electrolyte enables selective $Na^+$ transport along specific areas that are electronegative with sub-nanometre dimensions, resulting in a $Na^+$ conductivity of $1.30\times10^{-4}$ S cm$^{-1}$ and oxidative stability of up to 5.32 V (versus $Na^+$/Na) at $25\pm1\,°C$. Testing the quasi-solid-state electrolyte in $Na\|Na_3V_2(PO_4)_3$ coin cell configuration demonstrates fast reaction dynamics, low polarization voltages, and a stable cycling performance over 1000 cycles at 60 mA g$^{-1}$ and $25\pm1\,°C$ with a 0.0048% capacity decay per cycle and a final discharge capacity of 83.5 mAh g$^{-1}$.

Owing to the availability of Na and its physicochemical properties, sodium-ion batteries (SIBs) are regarded as a primary alternative for lithium-ion batteries (LIBs), especially in large-scale and distributed energy storage systems[1,2]. However, the high flammability and easy leakage of organic liquid electrolytes result in safety challenges for SIBs. The incorporation of solid-state electrolytes (SSEs), which exhibit electronic insulation properties but also high ionic conductivities, has been considered an effective strategy for tackling the high-temperature instabilities of non-aqueous alkali metal-based batteries[3]. Although inorganic solid-state electrolytes (ISSEs), such as Na superionic conductors (NASICON), sulfides, and complex hydrides, exhibit considerable ionic conductivity, good thermal stability, and robust mechanical strength, their intrinsic roughness and rigidity lead to poor contact with electrodes, resulting in high contact resistance and manufacturing difficulties[4–7]. Compared with ISSEs, solid polymer electrolytes (SPEs), such as poly(ethylene oxide) (PEO), polyethylene

glycol (PEG), and poly(vinylidene fluoride) (PVDF), can form a soft interface with electrodes through a good film-formation mechanism, improving interfacial stability[8–12]. However, the ionic conductivity of SPEs is highly dependent on their degree of amorphicity and lowest possible glass transition/melting temperatures ($T_g/T_m$)[11]. Thus, SPEs generally exhibit lower ionic conductivities (<10$^{-5}$ S cm$^{-1}$, 25 °C), low transference numbers (~0.2), and narrow electrochemical windows (<4 V), hindering their further commercialization in large-scale applications[13,14].

Quasi-solid-state polymer electrolytes, which are in an intermediate state between a liquid electrolyte and a solid electrolyte, simultaneously have a high room-temperature ionic conductivity and strong mechanical strength, and can provide stable working conditions for batteries, exhibiting great potential for practical application in solid-state sodium batteries (SSBs). As polymer hosts for quasi-solid-state electrolytes (QSSEs), covalent organic frameworks (COFs) with

[1]School of Material Science and Engineering, China University of Petroleum, 266580 Qingdao, China. ✉e-mail: liuzhengbeyond@163.com; yyg@upc.edu.cn; fanzhj666@163.com

periodic 1D channels, controllable pore structures, and functional groups have received attention in the energy storage field[15]. The permanent porosity of COFs can provide abundant free volume over a wide range of temperatures, thereby facilitating solid-state ion conduction at low temperatures (≤0 °C). In addition, the crystallographically fixed structure of COFs could provide homogeneously dispersed hopping sites while eliminating impedance stemming from electrolyte reorganization, which indicates that COFs are an ideal platform for Li+/Na+ conduction[16,17]. Although numerous COF-based QSSEs have been prepared through direct coordination or integration with cations, anions, or oxygen-containing functional groups[18–21], their Li+/Na+ conductivities are still unsatisfactory (<10$^{-5}$ S cm$^{-1}$, 25 °C) and are even worse (<10$^{-6}$) below −20 °C. Moreover, large amounts of plasticizers (>30 wt.%) are commonly added to ensure fast ion conduction and good electrode/electrolyte interfaces[22–24].

Recently, bionic technology inspired by biological systems has drawn increasing attention, which could promote the integration of function, intelligentization, and informatization of advanced materials. With regards to ion conductive properties, ion channels of biological membranes exhibit negatively charged (−COO$^-$)-modified inwalls and sub-nanometer structures[25–27], which can realize the selective and fast transport of Na+/K+ through the electrostatic repulsion of anions. Moreover, such a sub-nanometer space can confine solvent molecules in QSSEs, which can be beneficial for improving the electrode/electrolyte interface compatibility and reducing the interfacial impedance between particles. Inspired by this, the introduction of sub-nanometer pore size and (−COO$^-$)-modified skeleton in COFs can be considered a compelling opportunity to mimic the ion channels of biomembranes[28]. More significantly, due to the accurate modification of chemical structures at the atomic level, the QSSEs of COF-based biological ionic channels can act as an ideal platform for studying ion-conducting mechanisms.

Herein, inspired by the biological ion channels of the cell membrane, a COF-based QSSE membrane with biomimetic Na+ channels is prepared. The adjacent carboxylic acid groups (−COO$^-$) anchored on the COF inwalls form sub-nanochannels (6.7–11.6 Å), which can dissociate sodium bis(trifluoromethanesulfonyl)imide (NaTFSI) salt and boost the selective transport of Na+ (Fig. 1). Density functional theory (DFT) calculations and molecular dynamics (MD) simulations reveal that the localized distribution and fast conduction mechanism of Na+ are similar to those of the Na+/K+ channels of the cell membrane. Due to the electronegativity of −COO$^-$, Na+ is preferentially adsorbed in the sub-nanometer-sized zones, while TFSI$^-$ is repulsed at the center of the COF channels. Moreover, the addition of a propylene carbonate (PC) solvent can effectively wet the grain boundary and improve the electrolyte/electrode interface compatibility. Due to the sub-nanometer confinement effects promoted by the COFs, the solvent tends to attach to the interior of the COFs, making it harder to oxidize and decompose the QSSE, thus effectively increasing the Na+ conduction and the working temperature range. As a result, the COF-based QSSE displays a Na+ conductivity of $1.30 \times 10^{-4}$ S cm$^{-1}$ at 25 ± 1 °C and a larger Na+ transference number ($t_{Na+}$) of 0.90, revealing its potential for practical application in SSBs. When assembled and tested in Na||Na$_3$V$_2$(PO$_4$)$_3$ coin cell and single-layer pouch cell configurations, the COF-based QSSE facilitates efficient cycling performance in the specific current range of 12–60 mA g$^{-1}$ at 25 ± 1 °C.

## Results and discussion
### Material synthesis and characterization

A COF with a precisely designed pore structure and negatively charged modified inwalls was prepared according to a biomimetic design concept (Fig. 1 and Supplementary Fig. 1). The COF with high crystallinity and nanoscale pore structure (TPBD) was synthesized through a solvothermal method[29]. The powder X-ray diffraction (PXRD) pattern of TPBD shows a specific 2θ peak at a low angle of 3.4°, attributed to the (100) plane. The broad peak at a higher 2θ degree (~27°) corresponds to the π-π stacking effect of the (001) plane between COF layers, and the $d$ spacing is calculated to be ~3.29 Å (Supplementary Fig. 2). To mimic the negative inwalls of biological Na+ channels, the negative (−COOH)-modified COF (TPDBD) was modified by replacing the benzidine (BD) with a 4,4′-diamino-[1,1′-biphenyl]−2,2′-dicarboxylic acid (DBD) monomer. To further increase the negativity of the COF inwalls, the H+ of −COOH was exchanged with Na+ through a Na-activation process with 1 mM NaOH, and the sample was defined as "TPDBD-CNa". Due to the increased repulsion between COF layers after the introduction of −COO$^-$ groups, the diffraction peak of the (001) plane of TPDBD-CNa decreases to ~26°, and the corresponding $d$ spacing increases to ~3.39 Å. Moreover, the −COO$^-$ groups anchored

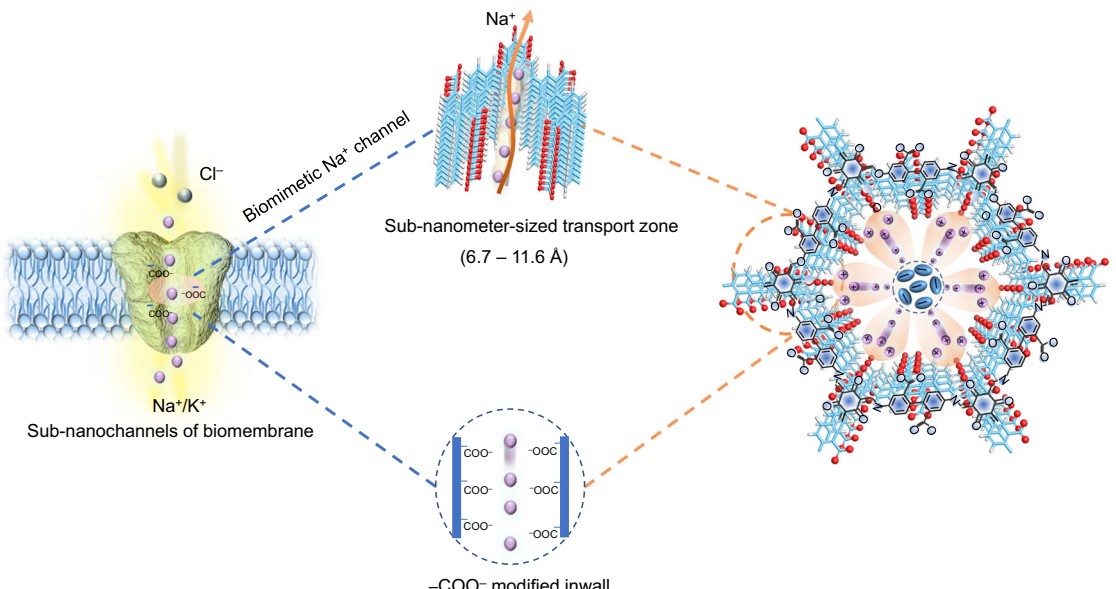

**Fig. 1 | Structure design.** Biomimetic concept of sub-nanometer-sized Na+ transport zones constructed by adjacent −COO$^-$ groups and COF inwalls (the purple and red spheres and cyan ovals denote sodium and oxygen and TFSI$^-$, respectively, and the cyan sticks and molecular structures denote the covalent organic framework).

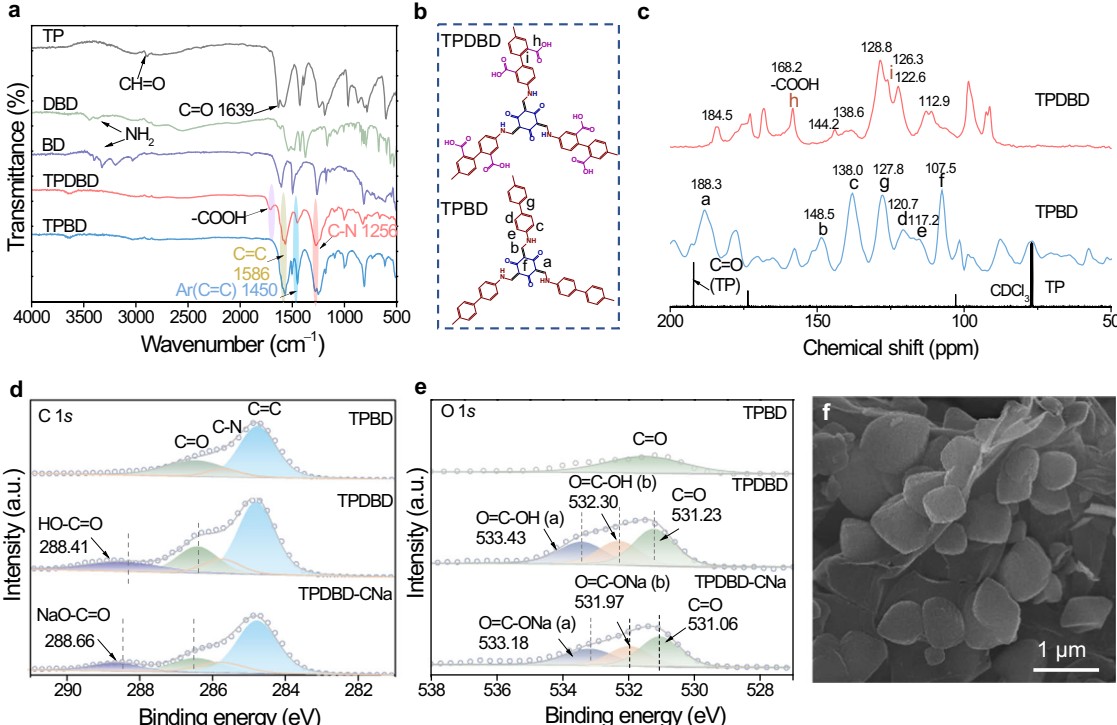

**Fig. 2 | Physical characterization of the COFs. a** FTIR spectra of TPBD, TPDBD, BD, DBD, and TP. **b** Partial unit structures of TPBD and TPDBD. **c** $^{13}$C CP-MAS solid-state NMR spectra of TPBD, TPDBD, and TP. **d** XPS C 1*s* spectra of TPBD, TPDBD, and TPDBD-CNa. **e** XPS O 1*s* spectra of TPBD, TPDBD, and TPDBD-CNa. **f** SEM image of TPDBD.

onto the COF have a large steric hindrance effect and disturb the π − π stacking interactions among the layers (Supplementary Fig. 3), corresponding with a wider interface distribution at a low angle[30]. Such lower crystallinity of TPDBD is conducive to fast ion transport through the effective elimination of grain boundaries[19,31–33]. The Brunauer–Emmett–Teller (BET) surface area and total pore volume of TPBD are 281 m$^2$ g$^{-1}$ and 0.42 cm$^3$ g$^{-1}$, respectively, which decrease to 74 m$^2$ g$^{-1}$ and 0.30 cm$^3$ g$^{-1}$ for TPDBD due to the introduction of −COO$^-$ groups (Supplementary Fig. 4). After Na-activation, the specific surface and pore volume of TPDBD-CNa increase again to 116 m$^2$ g$^{-1}$ and 0.60 cm$^3$ g$^{-1}$ due to the prevention of the formation of protic acid cross-linked oligomers inside the pores. Interestingly, the distances between the adjacent carbonyl functional groups anchored on the COF inwalls are measured in the range of 6.7–11.6 Å, which is similar to that of the anionic terminal-modified sub-nanostructure ion channels of the cell membrane (Supplementary Fig. 5)[26,27].

Compared with those of the BD and DBD monomers, the N-H stretching bands of free diamine (3100-3300 cm$^{-1}$) in the Fourier transform infrared (FTIR) spectra of TPBD, TPDBD, and TPDBD-CNa completely disappear, indicating the complete reaction of amino groups (Fig. 2a and Supplementary Fig. 6). Moreover, the carbonyl (C = O) peak at 1612 cm$^{-1}$ (1,3,5-triformylphloroglucinol (TP) monomer: 1639 cm$^{-1}$) broadens, and a C = C peak at 1586 cm$^{-1}$, a C = C aromatic ring (Ar) peak at 1450 cm$^{-1}$, and a peak of the C − N bond of the linker at 1256 cm$^{-1}$ appear, confirming the formation of a COF. In addition, a new specific C = O stretching band peak at 1711 cm$^{-1}$ (carboxylic acid) appears for TPDBD and TPDBD-CNa, indicating the successful incorporation of −COO$^-$ groups[33]. As shown in the $^{13}$C cross-polarization magic-angle-spinning (CP-MAS) solid-state nuclear magnetic resonance (NMR) spectra, after the Schiff base reaction, the specific shift of the C = O peak (~192 ppm) of the TP monomer completely disappears, and peaks of the keto carbonyl carbon (a) (~188.3 ppm for TPBD and ~184.5 ppm for TPDBD) and biphenyl junction (g) (~127.8 for TPBD and ~128.8 ppm for TPDBD) appear (Fig. 2b, c). In addition, chemical

shifts of the C = O (h) (~168.2 ppm) and C (i) (~126.3 ppm) peaks are observed for TPDBD, further indicating the successful incorporation of −COOH groups[21,29]. The Raman spectra display distinct scattering bands at ~1200, 1386, and 1598 cm$^{-1}$, indicating the retention of the structural rigidity of the COF throughout the synthesis process (Supplementary Fig. 7).

X-ray photoelectron spectroscopy (XPS) was performed to analyze the surface chemical states of TPBD, TPDBD, and TPDBD-CNa (Fig. 2d, e). As shown in the C 1*s* and O 1*s* XPS spectra of TPBD, the C = O peak is attributed to the keto carbonyl group, indicating the successful synthesis of the COF-TPBD framework. For TPDBD, specific C = O (286.41 eV) and HO−C = O (288.41 eV) peaks in the C 1*s* spectrum and O = C−OH (a, 533.43 eV), O = C−OH (b, 532.30 eV), and C = O (531.23 eV) peaks in the O 1*s* spectrum are observed[34]. Moreover, the O content of TPDBD is higher than that of TPBD (Supplementary Table 1), demonstrating the successful incorporation of carboxylic acid (−COOH) groups in the TPDBD framework. Similarly, the binding energies of C = O (286.48 eV) and NaO−C = O (288.66 eV) in the C 1*s* spectrum and those of O = C−ONa (a, 533.18 eV), O = C−ONa (b, 531.97 eV), and C = O (531.06 eV) in the O 1*s* spectrum, as well as the Na 1*s* peak at 1071.64 eV, indicate the successful exchange of H$^+$ with Na$^+$ in TPDBD-CNa (Supplementary Fig. 8). The scanning electron microscopy (SEM) and transmission electron microscopy (TEM) images of TPDBD show stacked nanosheets with micrometric and submicrometric morphologies (Fig. 2f and Supplementary Figs. 9 and 10), which is beneficial for interfacial contact and compatibility between electrodes and electrolyte[35]. The energy dispersive spectroscopy (EDS) images show that C, N, O, and Na are uniformly distributed in TPBD, TPDBD, and TPDBD-CNa (Supplementary Figs. 11–13).

The thermogravimetric analysis (TGA) profile of TPDBD shows a decomposition temperature of 274 °C (43% weight loss). The first stage of weight loss of TPDBD-CNa is mainly caused by the decomposition of −COO$^-$ at 264 °C, and the second stage begins at ~308 °C due to the collapse of the framework (46% weight loss). The addition of NaTFSI

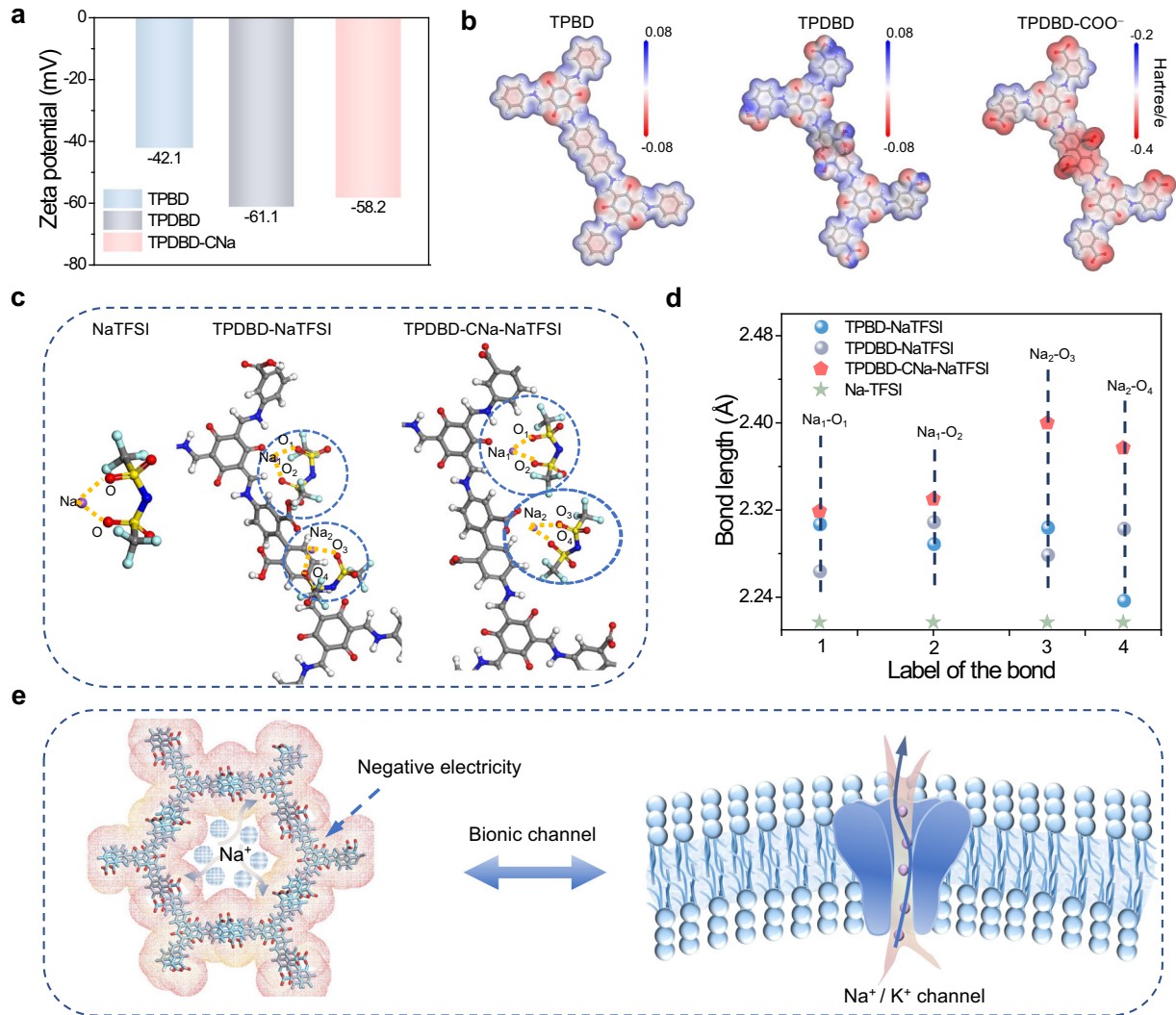

**Fig. 3 | Biomimetic Na⁺ channels. a** Zeta potential values of various COFs. **b** Electrostatic potential mappings of TPBD, TPDBD, and TPDBD-COO⁻ (the white, blue, gray, and red spheres denote hydrogen, nitrogen, carbon, and oxygen, respectively). **c** Optimized coordination structures between NaTFSI and TPDBD and TPDBD-COO⁻ (the white, blue, purple, gray, and red spheres denote hydrogen, nitrogen, sodium, carbon, and oxygen, respectively). **d** Bond lengths between NaTFSI and TPBD, TPDBD and TPDBD-COO⁻. **e** Schematic illustration of the bionic channel in TPDBD-CNa (the red spheres denote oxygen, and cyan sticks denote the covalent organic framework).

improves the thermal stability, and TPDBD-CNa-NaTFSI and TPBD-NaTFSI exhibit decomposition at ~380 and 410 °C with 62 and 60% weight loss up to 780 °C, respectively (Supplementary Fig. 14). Moreover, the specific surface and pore sizes of TPDBD-CNa-NaTFSI and TPBD-NaTFSI decrease to 18 m² g⁻¹ and 0.07 cm³ g⁻¹, 18 m² g⁻¹ and 0.08 cm³ g⁻¹ (Supplementary Fig. 15). The PXRD patterns of TPBD-NaTFSI and TPDBD-CNa-NaTFSI display the strong characteristic peaks of NaTFSI and weak small-angle diffraction peaks, indicating the efficient incorporation of Na salt into the COF channels (Supplementary Fig. 16)[36,37]. The FTIR and Raman spectra of TPBD-NaTFSI and TPDBD-CNa-NaTFSI further confirm the presence of NaTFSI (Supplementary Figs. 17 and 18). A blue shift in TPBD-NaTFSI and TPDBD-CNa-NaTFSI Raman spectra is observed compared with that of the NaTFSI solution, indicating the formation of cation/anion associations and fast Na⁺ transport channels in COF-NaTFSI[37–39]. The $^{23}$Na MAS NMR spectra of TPDBD-CNa-NaTFSI and TPBD-NaTFSI each exhibit two specific signals centered at −21.67 ppm (peak 1) and −42.36 ppm (peak 2) and −30.77 ppm (peak 1) and −42.39 ppm (peak 2), corresponding to Na⁺ in NaTFSI and Na-coordinated COFs (Supplementary Fig. 19). Moreover, elemental analysis (EA) and inductive coupled plasma (ICP) emission spectroscopy measurements verify that the Na contents of TPDBD-CNa-NaTFSI and TPBD-NaTFSI are 5.46 wt.% and 3.07 wt.%, respectively (Supplementary Table 1). Further EDS elemental mappings illustrate the successful introduction of Na salt into the COFs (Supplementary Fig. 20).

**Biomimetic Na⁺ channels**
Due to the negative inwalls decorated with −COO⁻ groups, the zeta potentials of TPDBD and TPDBD-CNa are −61.1 and −58.2 mV, much lower than that of TPBD (−42.1 mV), which is beneficial for the separation of Na⁺ and TFSI⁻ (Fig. 3a and Supplementary Fig. 21). The dissociation of NaTFSI in COFs is also confirmed by DFT calculations (Fig. 3b). As shown in the charge distribution mappings, the C = O and −COO⁻ groups of TPDBD and TPDBD-COO⁻ show a dark red color distribution, indicating more electronegative sites, which is advantageous for Na⁺ adsorption. In the optimized coordination structure, the bond length between Na⁺ and TFSI⁻ increases after the introduction of −COO⁻, indicating the dissociation of Na⁺ and TFSI⁻ (Fig. 3c, d and Supplementary Fig. 22). The negatively charged TPDBD-CNa inwalls can effectively capture Na⁺ and electrostatically repulse TFSI⁻,

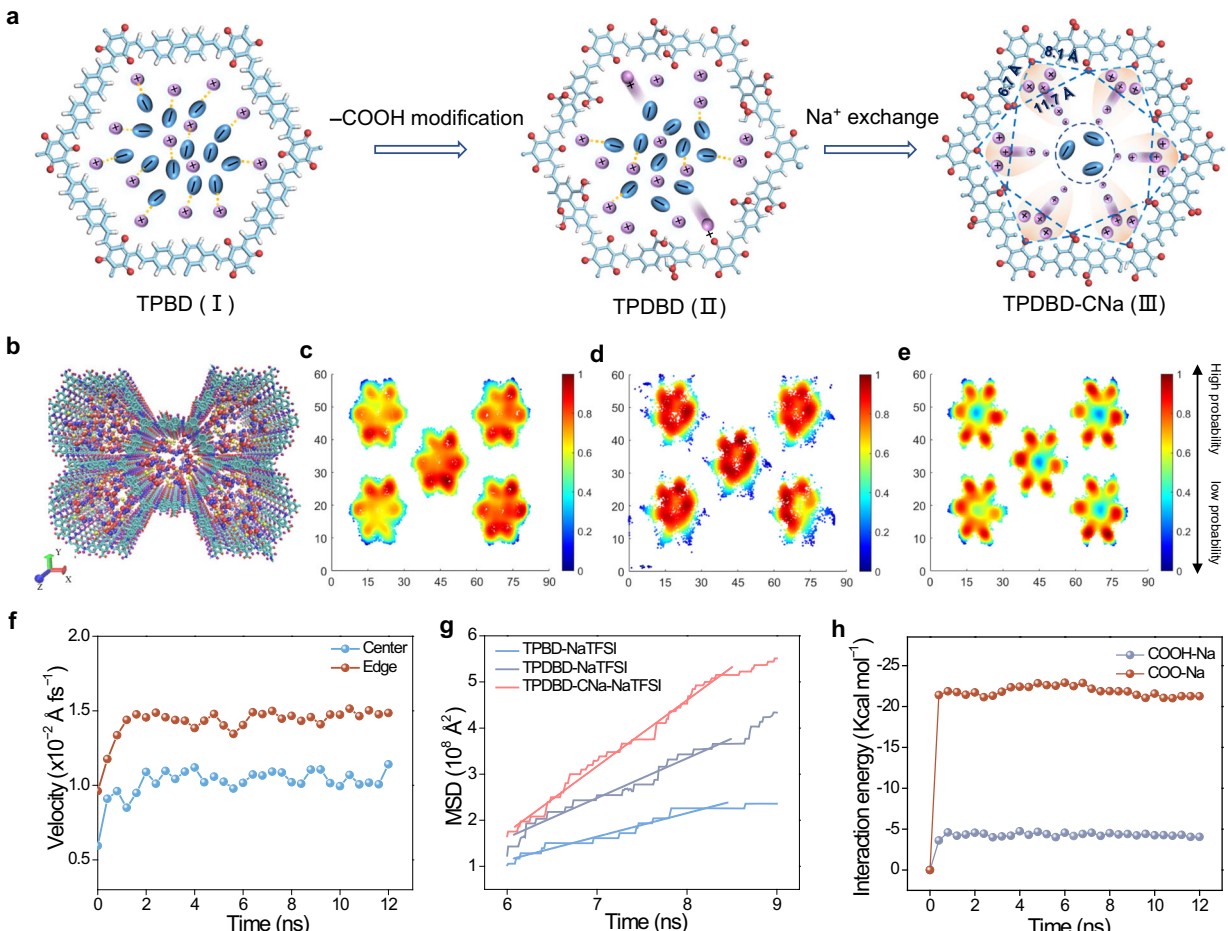

**Fig. 4 | Na⁺ transport mechanism of COF-NaTFSI. a** Distribution modes of Na⁺ and TFSI⁻ in TPBD, TPDBD, and TPDBD-CNa (the purple and red spheres and cyan ovals denote sodium, oxygen, and TFSI⁻, respectively, and the cyan sticks denote the covalent organic framework). **b** Schematic of the MD simulations of TPDBD-CNa-NaTFSI (the white, blue, cyan, pink, yellow, purple, and red spheres denote hydrogen, nitrogen, carbon, fluorine, sulfur, sodium, and oxygen, respectively).

Na⁺ density mappings (the red regions indicate the highest probability of Na⁺) of TPBD-NaTFSI (**c**), TPDBD-NaTFSI (**d**), and TPDBD-CNa-NaTFSI (**e**). **f** Na⁺ velocities at the edge and center of the TPDBD channel. **g** MSD results of Na⁺ over time in three kinds of COFs. **h** Interaction energy between −COOH/−COO⁻ groups and Na⁺ in TPDBD.

consequently facilitating rapid Na⁺ transfer when applied in QSSEs. It is worth noting that the sub-nanostructured ion channels with more negatively charged sites (−COO⁻) are similar to the biomimetic Na channels of the cell membrane (Fig. 3e).

MD simulations were used to understand the Na⁺ transport mechanism of the biomimetic COF with sub-nanometer-sized zones constructed by the adjacent −COO⁻ groups and COF inwalls. The distribution modes of Na⁺ and TFSI⁻ in TPBD, TPDBD, and TPDBD-CNa are shown in Fig. 4a. Schematic diagrams of COF-NaTFSI are shown in Fig. 4b and Supplementary Fig. 23. The Na⁺ density distribution maps of TPBD-NaTFSI show that most Na ions are uniformly distributed across the entire pore, and few Na ions are adsorbed on carbonyl groups derived from enol-to-keto tautomerism (Fig. 4c). After the introduction of −COOH, additional Na ions are adsorbed onto the negatively charged inwalls of TPDBD-NaTFSI (Fig. 4d). After the Na activation of TPDBD-CNa-NaTFSI, sub-nanoscale zones constructed by the adjacent −COO⁻ groups and COF inwalls are observed. Na ions are centrally distributed at the edge zones, while the TFSI⁻ anions are repulsively confined into the center of the COF channels (Fig. 4e). Each independent sub-nanometer zone, as in the Na⁺ channel of the cell membrane, is beneficial for selective Na⁺ transport. Indeed, the Na⁺ diffusion velocity at the edge zones is higher than that in the center sites due to the construction of fast Na⁺ migration pathways, as in the ion channels of the cell membrane (Fig. 4f). Figure 4g shows the mean

square displacement (MSD) of Na⁺ over time in three kinds of structures. The self-diffusion coefficients ($D_{TPBD-NaTFSI}$ ~ 0.085, $D_{TPDBD-NaTFSI}$ ~ 0.143, and $D_{TPDBD-CNa-NaTFSI}$ ~ 0.239 ($10^8$ Å² ns⁻¹)) were calculated from the slope of the MSD averaged over the trajectories of individual Na⁺ particles[40]. The largest $D_{TPDBD-CNa-NaTFSI}$, indicating the highest Na⁺ transport efficiency, was calculated for TPDBD-CNa, suggesting that the (−COO⁻)-based biomimetic channels can increase Na⁺ transport efficiency along the sub-nanometer-sized zones. The interaction energy between −COO⁻ (−4 kcal mol⁻¹)/−COOH (−22 kcal mol⁻¹) and Na⁺ in the COF was calculated, and the largest interaction energy between carbonyl groups and Na⁺ promotes the adsorption of Na⁺ in the sub-nanometer zones (Fig. 4h). The accurate distribution of charge and space around COF inwalls endows the biomimetic channel with high Na⁺ conduction performance.

## Na⁺ conduction properties

A large grain boundary impedance can hinder the practical ion conductivity and the high-efficiency utilization of solvent in QSSEs, which are essential aspects for producing high-performance Na-based cells. As shown in Supplementary Fig. 24, a small amount of PC solvent can be rapidly adsorbed into the COF membrane. The thermal stability of TPBD-NaTFSI and TPDBD-CNa-NaTFSI with a small amount of PC additive was investigated by TGA (Supplementary Fig. 25). Due to the confinement effects of the COF, ~9 wt.% PC solvents tend to attach to

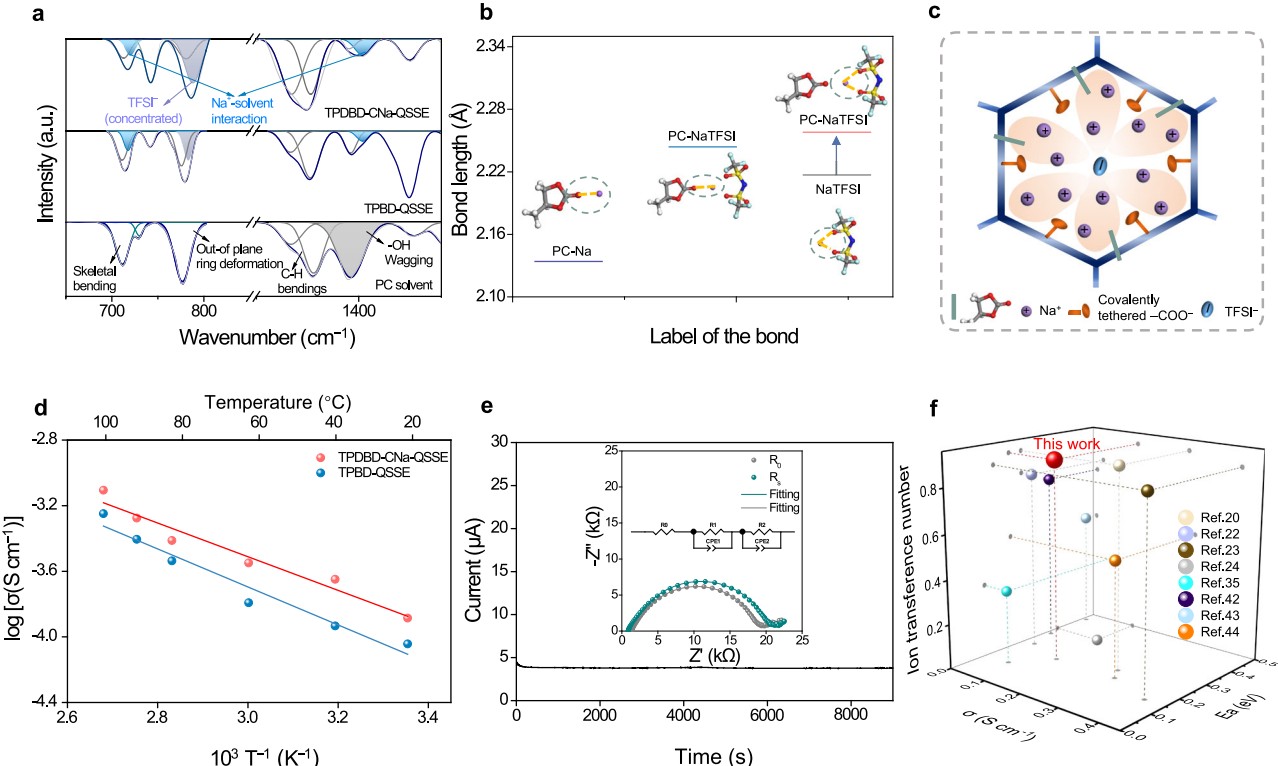

**Fig. 5 | Physicochemical properties and Na⁺ conduction of COF-QSSEs. a** FTIR spectra of the prepared TPDBD-CNa-QSSE, TPBD-QSSE, and PC solvent (cycled COF films in Na||Na cells over 20 cycles at 0.02 mA cm⁻² and 25 ± 1 °C). **b** Na−O bond lengths in PC-Na and PC-NaTFSI. **c** Schematic diagram of the PC, Na⁺, and TFSI⁻ distributions in TPDBD-CNa (the white, blue, gray, cyan, yellow, purple, and red spheres denote hydrogen, nitrogen, carbon, fluorine, sulfur, sodium, and oxygen,

respectively). **d** Arrhenius plot of the ionic conductivity of TPDBD-CNa-QSSE and TPBD-QSSE. **e** Current-time curves of the Na|TPDBD-CNa-QSSE|Na symmetric cell (inset shows the EIS at initial and steady states and corresponding equivalent circuit model). **f** Comparison of the typical performance of reported Li⁺/Na⁺ SSEs (the numerical values and testing temperature for the ionic conductivities provided in Supplementary Table 4).

the interior of the COF and decompose more slowly than pure solvent and glass fiber/solvent. MD simulation shows that the PC molecules are uniformly adsorbed and infiltrated on the TPDBD-CNa skeletons and aggregated on the TPBD surface, indicating that the well-designed biomimetic COF is beneficial for the uniform wettability of PC on the grain boundaries (Supplementary Fig. 26a, b). The high-efficiency utilization of PC can improve the electrode/electrolyte interface compatibility and reduce the interfacial impedance between particles. Under the electric field, along with Na⁺ transport, some PC molecules are confined into the sub-nanochannels due to the high interaction energy between TPDBD-CNa and the PC solvent, which can effectively prevent PC volatilization and further promote Na⁺ conductivity (Supplementary Fig. 26c–e and Supplementary Note 1). The configurations of the corresponding QSSEs after 20 cycles in a Na|QSSE|Na symmetric cell were measured by FTIR spectroscopy (Fig. 5a). The TPDBD-CNa-QSSE exhibits stronger PC-Na⁺ interactions (720 and 1403 cm⁻¹) and more concentrated TFSI⁻ (787 cm⁻¹) than the TPBD-QSSE, demonstrating a stronger electrolyte aggregation effect[41]. As shown in the optimized configuration (Fig. 5b), the Na−O bond length of NaTFSI in the QSSEs becomes longer, indicating that PC can effectively dissociate Na⁺ and TFSI⁻. Figure 5c exhibits a schematic diagram of the PC, Na⁺, and TFSI⁻ distributions in the QSSEs.

Electrochemical impedance spectroscopy (EIS) measurements of pellets were carried out from −40 to 100 °C to study the Na⁺ conduction properties of the QSSEs. The Ti|TPDBD-CNa-NaTFSI|Ti symmetric cell without solvent cannot work properly below 100 °C due to the weak contact between the COF and titanium sheet (Supplementary Fig. 27). The Na⁺ conductivity of the TPDBD-CNa-QSSE is 1.30 × 10⁻⁴ S cm⁻¹ at 25 ± 1 °C, higher than that of the TPBD-QSSE (9.06 × 10⁻⁵ S cm⁻¹, 25 ± 1 °C), and even at a low temperature of −40 °C,

the TPDBD-CNa-NaTFSI can still deliver an ionic conductivity of 8.98 × 10⁻⁶ S cm⁻¹ (Fig. 5d, Supplementary Figs. 28 and 29, Supplementary Table 2, and Supplementary Note 2). The tendency of increasing conductivity is consistent with the self-diffusion coefficients measured by MSD calculations (Fig. 4g). As the temperature increases, the Arrhenius plots exhibit a proportional increase in the logarithmic ionic conductivity, and the corresponding activation energy (Ea) values of the TPDBD-CNa-QSSE and TPBD-QSSE are 0.204 eV and 0.230 eV, respectively. Benefiting from the precise charge design of the biomimetic Na⁺ channel, the TPDBD-CNa-QSSE shows a larger $t_{Na+}$ of 0.90 than the TPBD-QSSE (0.74), suggesting a better ion migration capability (Fig. 5e, Supplementary Fig. 30, Supplementary Table 3, and Supplementary Note 3). The ion transport performances reported are well-positioned compared to the SSE state-of-the-art (Fig. 5f and Supplementary Tables 4 and 5)[20,22–24,35,42–44]. Such sub-nanometer-sized zones can be regarded as biomimetic Na⁺ transfer channels and can effectively promote the Na⁺ transport efficiency directionally along the oxygen atoms. Moreover, solvent molecules can be confined in the sub-nanoscale channels in QSSEs to reduce volatilization. The jumping transfer mode of Na⁺ from one carboxylic/carbonyl position to the next unoccupied site under a given voltage can be regarded as a kind of pendular Na⁺ transport mechanism, resulting in a high Na⁺ conduction performance (Supplementary Fig. 31).

Regarding the linear sweep voltammetry (LSV) measurement, an electrochemical stability window of 5.32 V was observed for the TPDBD-CNa-QSSE, which is wider than that of the TPBD-QSSE (4.39 V, Fig. 6a), indicating that the biomimetic Na⁺ channels with −COO⁻ groups endow the QSSE with a stronger antioxidant property and higher decomposition potential. To reveal the stability of the QSSE at the molecular level, the optimized structures and highest-occupied/

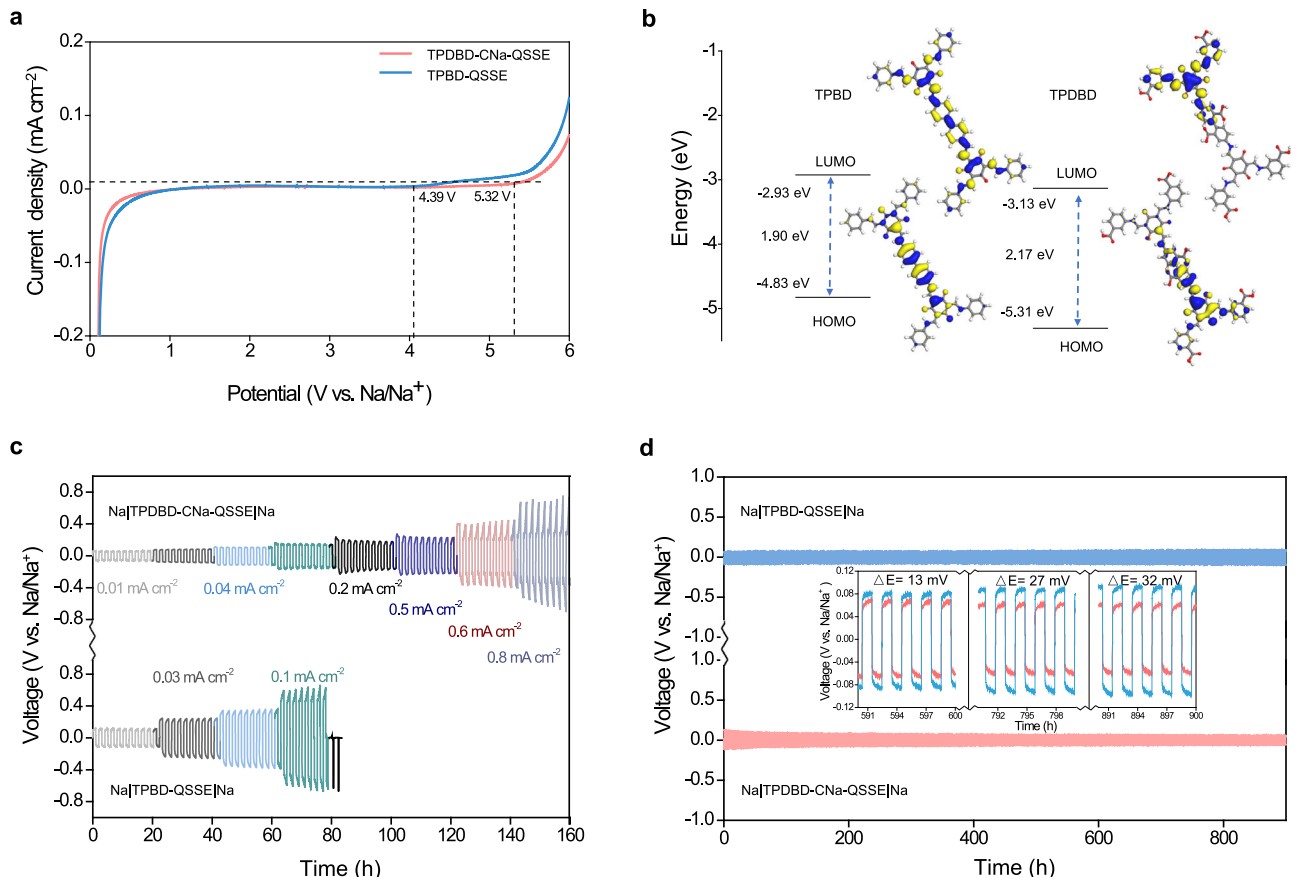

**Fig. 6 | Electrochemical window and Na plating/stripping of COF-QSSEs. a** LSV profiles of SS|TPBD-QSSE|Na and SS|TPDBD-CNa-QSSE|Na asymmetric cells. **b** Calculated HOMO and LUMO values of TPBD and TPDBD based on DFT calculations. **c** Rate performance of Na plating/stripping for Na|QSSE|Na at 25 ± 1 °C and 0.01, 0.03, 0.04, 0.1, 0.2, 0.5, 0.6, and 0.8 mA cm⁻². **d** Na plating/stripping of Na| QSSE|Na at a current density of 0.01 mA cm⁻² and 25 ± 1 °C for 900 h and 2 h per cycle (the inset shows the time-division plating/stripping curves).

lowest unoccupied molecular orbitals (HOMO/LUMO) of TPBD and TPDBD were determined by DFT calculations (Fig. 6b). It should be noted that a lower HOMO energy indicates that it is harder to lose electrons, corresponding to a higher oxidation potential. Due to the introduction of electron-withdrawing –COOH groups, TPDBD shows a lower HOMO energy (−5.31 eV) than TPBD (−4.83 eV), indicating the stronger antioxidant property of the TPDBD-CNa-QSSE[36,45]. Moreover, the band gap ($E_g$) of TPDBD is determined to be 2.17 eV, larger than that of TPBD (1.90 eV), further implying that the COF modified with –COO⁻ groups has better electron insulation, which is preferred for QSSEs. Moreover, UV–vis absorption spectra (Supplementary Fig. 32) were used to test the absorption edge and band gap of TPBD (2.15 eV), TPBD-NaTFSI (2.12 eV), TPDBD (2.38 eV), TPDBD-CNa (2.35 eV), and TPDBD-CNa-NaTFSI (2.34 eV)[46]. The changing trend observed in the UV–vis spectra (band gap order: TPBD < TPDBD) is consistent with the DFT calculations.

Na plating/stripping plots of the Na|QSSE|Na symmetric cells are displayed in Fig. 6c. Approximately 9 wt.% solvent (PC with 5% fluoroethylene carbonate (FEC)) was added to infiltrate the interface, as FEC can passivate the Na surface and improve the interface stability. Compared with TPBD-QSSE (unstable above 0.1 mA cm⁻²), TPDBD-CNa-QSSE exhibits lower polarization and more stable Na plating/stripping without any sign of short-circuiting. The polarization voltages are ±50, ±75, ±110, ±150, ±180, ±224, ± 450, and ±700 mV at 0.01, 0.03, 0.04, 0.10, 0.2, 0.5, 0.6, and 0.8 mA cm⁻², respectively. Figure 6d describes the long-term interfacial stability between the electrolyte and Na metal electrode using the Na||Na symmetric cell at 0.01 mA cm⁻² for 2 h per cycle. The TPBD-QSSE

and TPDBD-CNa-QSSE both exhibit stable Na plating/stripping behavior over 900 h without any irreversible fluctuation of overpotential. The TPDBD-CNa-QSSE shows a lower voltage plateau of ~62 mV compared to the TPBD-QSSE (from 73 to 93 mV from 400 to 900 h). The symmetric cell with the TPDBD-CNa-QSSE at 0.05 mA cm⁻² exhibits steady Na insertion/extraction processes for over 450 h without obvious fluctuation of potential (Supplementary Fig. 33). To verify the practicality of the prepared QSSEs at low temperatures, Na plating/stripping at 0, −5, −10, −15, −20, and −25 °C and symmetric cell operations at 0 °C were carried out (Supplementary Fig. 34a). As the temperature is reduced from 0 to −20 °C, the Na|TPDBD-CNa-QSSE|Na symmetric cell shows low overpotential and stable voltage profiles under 10 µA cm⁻². When the temperature is further reduced to −25 °C, irregular voltage fluctuations emerge, which can be ascribed to unstable Na plating/stripping and the formation of dead Na (i.e., Na metal regions that are electronically disconnected from the current collector)[47,48]. However, due to the absence of biomimetic sub-nanochannels, the voltage irregularity of the Na|TPBD-QSSE|Na symmetrical cell appears at a relatively low temperature of −10 °C. Moreover, Supplementary Fig. 34b shows that relatively stable voltage curves were obtained for over 550 h without overpotential fluctuation at 0 °C, further indicating that the bioinspired ionic channel design is beneficial for uniform Na⁺ deposition/stripping. A small amount of PC additive was confined inside the TPDBD-CNa-QSSE, alleviating interface side reactions caused by self-decomposition and thus facilitating more stable and higher-current-density Na plating/stripping cycles.

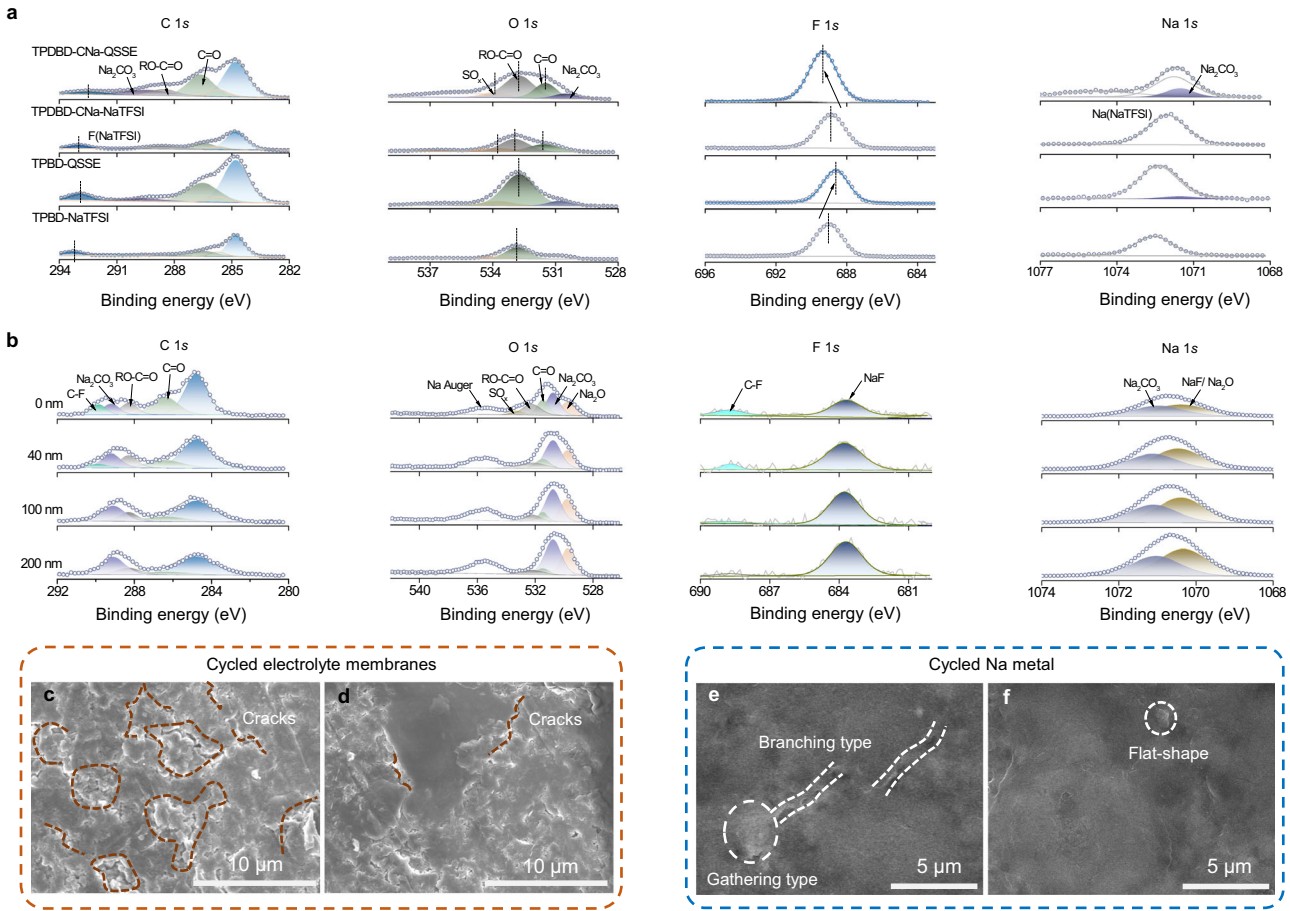

**Fig. 7 | Ex-situ postmortem XPS and SEM characterizations. a** XPS C 1*s*, O 1*s*, F 1*s*, and Na 1*s* spectra of the TPDBD-CNa-QSSE/TPBD-QSSE in Na∥Na cells over 20 cycles at 0.02 mA cm$^{-2}$ and 25 ± 1 °C and TPDBD-CNa-NaTFSI/TPBD-NaTFSI. **b** XPS C 1*s*, O 1*s*, F 1*s*, and Na 1*s* depth profiles of cycled Na metal over 20 cycles at 0.02 mA cm$^{-2}$ and 25 ± 1 °C in a Na|TPDBD-CNa-QSSE|Na symmetric cell. SEM images of cycled membranes of the TPBD-QSSE (**c**) and TPDBD-CNa-QSSE (**d**) over 20 cycles at 0.02 mA cm$^{-2}$ and 25 ± 1 °C. SEM images of the cycled Na metal of the TPBD-QSSE (**e**) and TPDBD-CNa-QSSE (**f**) over 20 cycles at 0.02 mA cm$^{-2}$ and 25 ± 1 °C.

XPS experiments were carried out to study the chemical composition of the solid electrolyte interphases (SEIs) formed on the surfaces of the cycled COF-QSSEs and Na electrodes (Fig. 7a, b). The C 1*s*, O 1*s*, F 1*s*, and Na 1*s* spectra were used to estimate the coexistence of organic species (C = C/C − C, C = O, C − F) and inorganic species (Na$_2$CO$_3$, Na$_2$O, and NaF) in the SEI layers. As shown in Fig. 7a, compared with those for the TPBD-QSSE, the specific peaks of inorganic species increase, and those of organic species decrease slightly for the TPDBD-CNa-QSSE. The inorganic species are beneficial for the construction of stable and compact SEI layers, which is conducive to the inhibition of Na dendrite growth. Furthermore, compared with those of TPDBD-CNa-NaTFSI, the binding energies of the RO − C = O and C = O(Na) peaks of TPDBD-CNa-QSSE decrease while the binding energy of the F 1*s* peak increases. These results further confirm that Na$^+$ tends to be distributed in sub-nanometer-sized zones, while TFSI$^-$ is electrostatically repulsed in the central region, which is consistent with the MD simulations. To gain more insights into the SEI composition, the cycled Na anode surface composition of the Na|TPDBD-CNa-QSSE|Na symmetrical cell was further analyzed by conducting in-depth Ar$^+$ sputtering at 0, 40, 100, and 200 nm. As shown in the elemental content distribution at different SEI depths (Supplementary Fig. 35), C and O are the dominant elements on the surface, and the atomic percentage of C undergoes a sharper decrease than that of O upon sputtering. As the sputtering depth increases, the atomic percentage of Na and the Na/C ratio gradually increase, indicating a gradual approach towards the Na anode surface. The C 1*s*, O 1*s*, F 1*s*, and Na 1*s* spectra were used to study the chemical

composition of the SEI at different depths (Fig. 7b). As shown in the C 1*s* spectra, organic species, such as C − C, C = O, RO − C = O, and C − F species, and inorganic species, such as Na$_2$CO$_3$, with fractional contents are observed in the outer SEI layer. As the sputtering depth increases, the peak intensity of organic species decreases, and the peak intensity of inorganic species gradually increases. Such a variation trend is also observed in the O 1*s* spectra. When the sputtering depth is 100 nm, the C − F peak almost disappears, and the NaF signal becomes stronger, indicating that the dominant species are inorganic species, such as NaF and Na$_2$CO$_3$, in the inner layer of the SEI. It should be emphasized that C − F species can restrict Na$^+$ transport and NaF species can accelerate ion transport through the SEI[49,50]. Furthermore, the O and F element contents remain almost constant, indicating the homogeneous distribution of the SEI composition along with the various sputtering depths. In conclusion, the SEI film of the Na anode based on the TPDBD-CNa-QSSE is composed of fewer organic carbonates and contains dense inorganic substances; thus, the SEI has a high mechanical strength and is beneficial for the inhibition of dendrite growth[51,52]. The surface morphologies of the membrane and Na metal recovered after 20 cycles were investigated by SEM observations (Fig. 7c–f). The membrane of the cycled Na|TPBD-QSSE|Na cell exhibits more cracks than that of the Na|TPDBD-CNa-QSSE|Na cell. The accumulated stress/strain during Na stripping/plating processes can accelerate interface deterioration, resulting in rapid dendrite growth. Moreover, the branching and gathering of Na dendrites are observed in the Na|TPBD-QSSE|Na cell to a greater degree than in the Na|TPDBD-

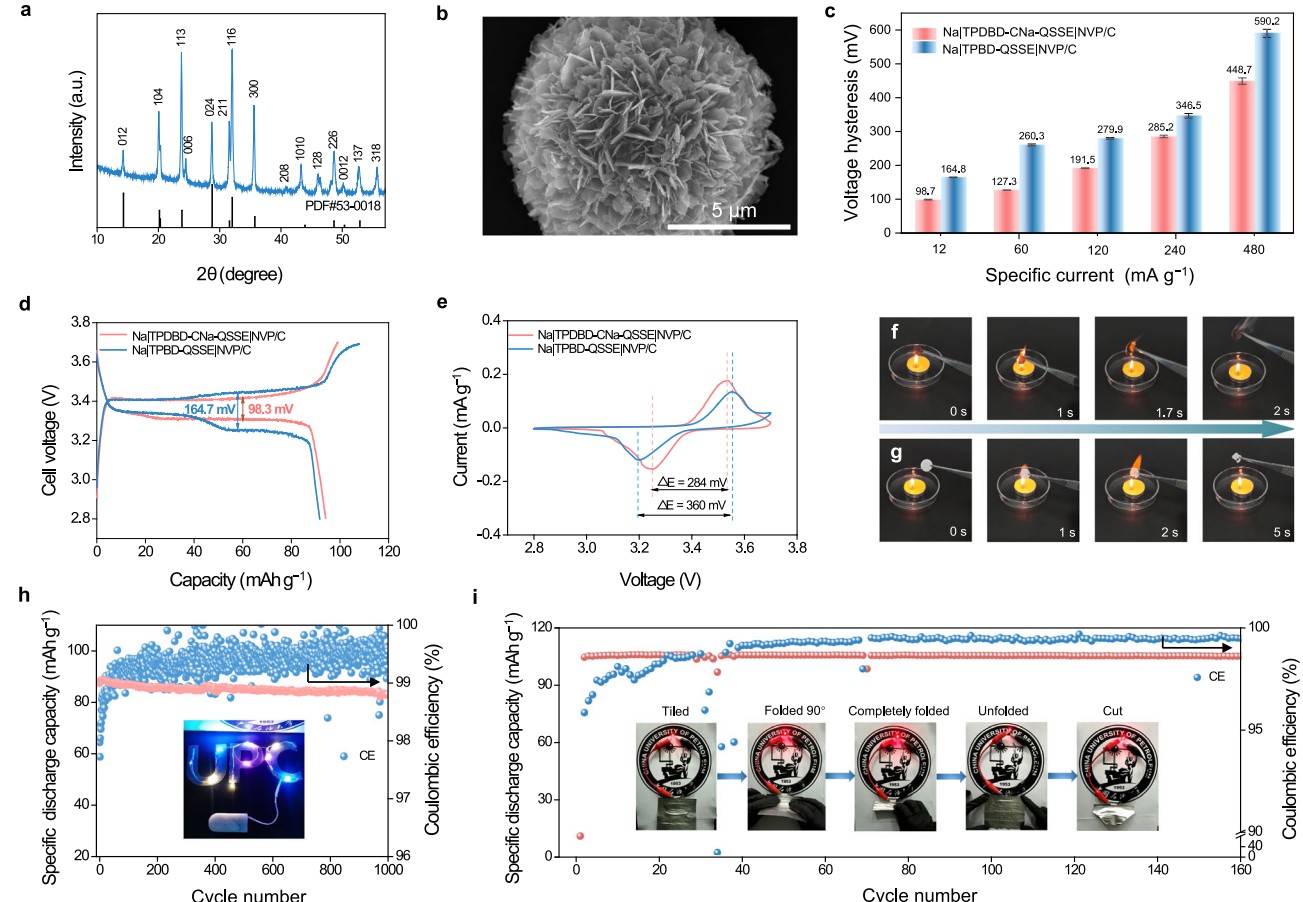

**Fig. 8 | Na|QSSE|NVP/C cell and safety properties. a** XRD pattern of NVP/C. **b** SEM image of NVP/C. **c** Histograms of the voltage hysteresis under different specific currents (the error bar represents the standard deviation of the average voltage of charge/discharge near 50 mAh g⁻¹). **d** Charge/discharge voltage profiles of the Na| TPDBD-CNa-QSSE|NVP/C and Na|TPBD-QSSE|NVP/C cells at 12 mA g⁻¹. **e** CV curves of the Na|TPDBD-CNa-QSSE|NVP/C and Na|TPBD-QSSE|NVP/C cells at 0.1 mV s⁻¹. **f** Combustion behaviors of the TPDBD-CNa-QSSE. **g** Combustion behaviors of the solvent-infiltrated fiberglass separator. **h** Long-cycle stability of the Na|TPDBD-CNa-QSSE|NVP/C cell at 60 mA g⁻¹ (the inset shows a photo of ten 2.5 V LEDs). **i** Cycling performance of the Na|TPDBD-CNa-QSSE|NVP/C tiled pouch cell at 12 mA g⁻¹ for 160 cycles. The inset photos show a LED powered by the QSSE pouch cell in different states of "Tiled", "Folded 90°", "Completely folded", "Unfolded", and "Cut".

CNa-QSSE|Na cell. These results illustrate that the TPDBD-CNa-QSSE exhibits good interfacial compatibility with the Na anode and suppresses the growth of Na dendrites due to a stable voltage response.

**Assembly and testing of quasi-all-solid-state Na|QSSEs| $Na_3V_2(PO_4)_3$/C cells**

To prove the practical application of the QSSEs, Na|QSSEs| $Na_3V_2(PO_4)_3$/C cells were assembled using a carbon-doped $Na_3V_2(PO_4)_3$ (NVP/C) positive electrode, a sodium metal negative electrode, and the QSSEs (Supplementary Fig. 36). First, NVP/C with a highly crystalline phase (PDF: #53-0018) was prepared by a hydrothermal method (Fig. 8a)[53,54]. SEM images of NVP/C show a branched morphology consisting of nanometer flakes with an average thickness of ~40 nm. Such a 3D interconnected structure is beneficial for fast Na⁺ diffusion and relieves mechanical stress caused by Na⁺ insertion/extraction (Fig. 8b and Supplementary Fig. 37). TGA, Raman, and BET analyses confirm the successful introduction of a hierarchical porous carbon layer into the NVP/C structure (Supplementary Figs. 38–40).

The rate performances of the Na|TPDBD-CNa-QSSE|NVP/C and Na|TPBD-QSSE|NVP/C cells are shown in Supplementary Fig. 41. When the specific currents are 60, 120, and 240 mA g⁻¹, the specific capacities of the Na|TPDBD-CNa-QSSE|NVP/C cell are 91.8, 87.6, and 82.1 mAh g⁻¹, respectively. Even at a higher rate of 480 mA g⁻¹, the cell still delivers a capacity of 68.9 mAh g⁻¹, higher than that of the Na|

TPBD-QSSE|NVP/C cell (60.1 mAh g⁻¹ at 480 mA g⁻¹). When the specific current decreases to 60 mA g⁻¹, a specific capacity of 91.7 mAh g⁻¹ can be obtained, indicating better reversible characteristics compared to those of the Na|TPBD-QSSE|NVP/C cell (87.38 mAh g⁻¹ at 60 mA g⁻¹). The charge/discharge voltage profiles of the Na|TPBD-QSSE|NVP/C cell show polarization potentials of 164.8, 260.3, 279.9, 346.5, and 590.2 mV from 12 to 480 mA g⁻¹ (Fig. 8c, d and Supplementary Fig. 42). Benefiting from the fast conduction of Na⁺ and good interfacial stability between the QSSE and electrodes, the polarization voltages of the Na|TPDBD-CNa-QSSE|NVP/C cell are 98.7, 127.3, 191.5, 285.2, and 448.7 mV at specific currents of 12, 60, 120, 240 and 480 mA g⁻¹ at an average voltage of charge/discharge near 50 mAh g⁻¹, respectively. Such low electrochemical polarization of the Na|TPDBD-CNa-QSSE|NVP/C cell ensures the structural integrity of the interface and maintains a stable voltage plateau during the long-term charge/discharge process. As illustrated in the cyclic voltammetry (CV) profiles (Fig. 8e), a pair of well-defined redox peaks appeared for both electrodes, corresponding to the reversible transformation of V³⁺/V⁴⁺ with two Na extraction/insertions following the electrochemical reaction $Na_3V_2(PO_4)_3 \leftrightarrow NaV_2(PO_4)_3$[55]. The CV curves of the Na|TPDBD-CNa-QSSE|NVP/C cell show a smaller potential gap and larger current response than those of the Na|TPBD-QSSE|NVP/C cell, indicating faster Na⁺ diffusion kinetics enhanced by the higher ion/electron conduction. In addition, CV curves were measured at various sweep rates to evaluate the Na⁺ diffusion

kinetics of the cells (Supplementary Fig. 43 and Supplementary Note 4). Compared with TPBD, the Na|TPDBD-CNa-QSSE|NVP/C cell exhibits oxidation and reduction peaks that have a relatively small potential gap between them and improved Na$^+$ conduction in the process of changing the sweep speed. The $D_{Na}^+$ values of the Na| TPDBD-CNa-QSSE|NVP/C cell for peak 1 (anodic) and peak 2 (cathodic) are determined to be $5.77 \times 10^{-13}$ and $6.23 \times 10^{-13}$ cm$^2$ s$^{-1}$, respectively, which are higher than those of the Na|TPBD-QSSE|NVP/C cell $(2.77 \times 10^{-13}$ and $1.92 \times 10^{-13}$ cm$^2$ s$^{-1})$, comparable to those in most previously reported works[54,56]. Figure 8f, g shows the combustion behaviors of the COF-QSSEs. Once the ignited TPDBD-CNa-QSSE membrane is moved away from the fire, the flames quickly extinguish, while the fiberglass separator can be ignited and burn with a bright flame accompanied by volume reduction, demonstrating the adequate flame resistance property of the TPDBD-CNa-QSSE.

Benefiting from electrode/electrolyte interface compatibility, the Na|TPDBD-CNa-QSSE|NVP/C cell exhibits excellent long-cycle stability at a specific current of 60 mA g$^{-1}$, and 83.5 mAh g$^{-1}$ is retained after 1000 cycles, corresponding to only 0.0048% capacity decay per cycle (Fig. 8h), which is comparable to that of most reported SSE-based batteries (Supplementary Table 6). Even at a higher specific current of 120 mA g$^{-1}$, 74.1 mAh g$^{-1}$ is retained after 500 cycles, corresponding to 0.0195% capacity decay per cycle (Supplementary Fig. 44). The inset photograph (Fig. 8h) displays ten 2.5 V light-emitting diode (LED) lights powered by the Na|TPDBD-CNa-QSSE|NVP/C cell, and good brightness was maintained for 12 h. However, without sub-nanochannel confinement solvents, overcharge behavior is observed during the long-term cycling of the Na|TPBD-QSSE|NVP/C cell even at a relatively low specific current of 60 mA g$^{-1}$ (Supplementary Fig. 45), indicating an unstable interface between the TPBD-QSSE and the Na electrode. To further confirm the practical application of the prepared QSSEs, the cycling stability and rate performance of the Na|TPDBD-CNa-QSSE|NVP/C cell with high cathode mass loadings of active materials (2.6 mg cm$^{-2}$) and a thin Na anode (~100 μm) were carried out (Supplementary Fig. 46 and Supplementary Note 5). The good cycle stability and reversibility verify that the Na|TPDBD-CNa-QSSE|NVP/C cell with a high mass loading of the cathode shows good rate performance and long-cycle stability. It should be noted that the Na|TPDBD-CNa-QSSE|NVP/C cell at 0 °C retains 103.1 mAh g$^{-1}$ after 200 cycles at a specific current of 12 mA g$^{-1}$ with a coulombic efficiency of 99.4% (Supplementary Fig. 47), confirming the good cycling stability of the QSSE-based cell at decreased temperatures. To investigate the practical application of the soft-pack Na|TPDBD-CNa-QSSE|NVP/C cell in flexible electronic devices (Supplementary Fig. 48), a freestanding film with a large area, high flexibility, and mechanical strength was prepared and is shown in Supplementary Fig. 49. The bendable Na|TPDBD-CNa-QSSE|NVP/C pouch cell (Supplementary Fig. 50) demonstrates good cycling performance, retaining 105.6 mAh g$^{-1}$ after 160 cycles at a specific current of 12 mA g$^{-1}$ with a coulombic efficiency of 99.5% (Fig. 8i). The inset photographs show that the LED device can function under different bending conditions, verifying the flexibility and safety of the bendable pouch cell. In addition, the Na|TPDBD-CNa-QSSE|NVP/C cell deliver high specific energies (calculated on the mass of active material in the positive electrode) of 313.2, 302.0, 288.6, 263.8, and 215.1 Wh kg$^{-1}$ at 12, 60, 120, 240, and 480 A g$^{-1}$, respectively (Supplementary Fig. 51).

In summary, a COF-QSSE with a biomimetic Na$^+$ channel design was prepared through the construction of six sub-nanometer zones by the introduction of –COO$^-$ groups into the COF inwalls. Benefiting from the accurate size of the cavity and carbonyl binding sites, the solvents are confined in the biomimetic sub-nanochannels, and the COF-based QSSE exhibits a high Na$^+$ conductivity of $1.30 \times 10^{-4}$ S cm$^{-1}$ and oxidative stability of up to 5.32 V (versus Na$^+$/Na) at 25 ± 1 °C. DFT calculations and MD simulations revealed that Na$^+$ undergoes a pore wall adsorption phenomenon (highly centralized to the carbonyl group) in the sub-nanochannels, and the –COO$^-$ groups anchored on

the COF inwalls are beneficial for the rapid dissociation of NaTFSI. Moreover, the electrolyte/electrode interface in the Na plating/stripping experiment is stable for 900 h of cycling. When assembled with the NVP/C cathode, Na metal anode, and TPDBD-CNa-QSSE membrane, the SSB shows 0.0048% capacity decay per cycle over 1000 cycles.

## Methods

### Materials
1,3,5-Triformylphloroglucinol (TP, 98%), P-phenylenediamine (BD, 97%), 4,4'-diamino-[1,1'-biphenyl]–2,2'-dicarboxylic acid (DBD, 98%) and sodium bis(trifluoromethanesulfonyl)imide (NaTFSI, 98%) were purchased from Zhengzhou Alpha Chemical Co., Ltd. Dimethylacetamide, tetrahydrofuran, acetone, vanadium pentoxide (V$_2$O$_5$), oxalate dihydrate (H$_2$C$_2$O$_4$·2H$_2$O), glucose, and sodium phosphate monobasic monohydrate (NaH$_2$PO$_4$·H$_2$O) were of analytical grade (AR) and purchased from Sinopharm Chemical Reagent Co., Ltd. without further purification unless indicated. Dioxane (99.7%, with molecular sieves, stabilized with BHT (butylated hydroxytoluene), water ≤50 ppm (by K.F. (Karl Fischer)), MkSeal) and mesitylene (98%, with molecular sieves, water ≤50 ppm (by K.F.), MkSeal) were purchased from Shanghai Macklin Biochemical Technology Co., Ltd.

### Preparation of TPBD and TPDBD
TPBD was synthesized by the following method: TP (63 mg, 0.3 mmol), BD (82.9 mg, 0.45 mmol), and 0.5 mL of 3 M aqueous acetic acid were added into a Pyrex tube containing dioxane/mesitylene (1/1 (v/v)) and sonicated for 10 min (KQ-600DV)[29]. TPDBD was synthesized by the following method: TP (21.0 mg, 0.10 mmol), DBD (40.8 mg, 0.15 mmol), and 0.4 mL of 6 M aqueous acetic acid were added into a Pyrex tube containing dioxane/mesitylene (1/3 (v/v)) and sonicated for 10 min (KQ-600DV). The above mixtures were flash-frozen under liquid nitrogen and degassed by three freeze-pump-thaw cycles. After then, these tubes were sealed and heated at 120 °C for 3 days. The resultant precipitate was collected by filtration and washed with dimethylacetamide, tetrahydrofuran, and acetone. Finally, the TPBD and TPDBD powders were obtained after drying at 120 °C under vacuum overnight.

### Preparation of TPDBD-CNa
To further increase the electronegativity of the COF inwalls, the H$^+$ of the –COOH group (TPDBD, 60.0 mg) was subsequently exchanged with Na$^+$ through a Na activation process in 250 mL 1 mM NaOH for 10 h, and then TPDBD-CNa was obtained. The product was dried under vacuum at 120 °C overnight.

### Preparation of TPBD-NaTFSI and TPDBD-CNa-NaTFSI
The synthesized TPBD and TPDBD-CNa powders (100 mg) were stirred in 20 wt.% NaTFSI methanol (MeOH, 200 mL) solution at 25 ± 1 °C for 48 h. The solid product was obtained by centrifugation, washed with MeOH several times and dried at 120 °C in vacuum overnight. Then, NaTFSI solution (100 mg in 2 mL MeOH) was added to the mixture and stirred at 25 ± 1 °C for 12 h. Finally, the solvent was removed under vacuum at 25 ± 1 °C, and the final product was degassed under vacuum at 120 °C overnight and dented as TPBD-NaTFSI and TPDBD-CNa-NaTFSI.

### Synthesis of Na$_3$V$_2$(PO$_4$)$_3$/C (NVP/C)
First, V$_2$O$_5$ (1 mmol, 0.182 g) was added to 30 mL n-dimethylformamide (DMF) and stirred at 80 °C for 0.5 h. Then, NaH$_2$PO$_4$·H$_2$O (3 mmol, 0.414 g), H$_2$C$_2$O$_4$·2H$_2$O (3 mmol, 0.378 g), and 5 mL water were added to the above solution and stirred for 1 h. After that, the solution was transferred into a 50 mL Teflon-lined stainless steel autoclave and heated at 180 °C for 20 h. Then, the product was collected by filtration and added to the DMF solution at a molar ratio of Na$_3$V$_2$(PO$_4$)$_3$: glucose

of 1:1. The solution was stirred for 0.5 h at 80 °C, and the as-prepared solution was dried in an oven overnight. Finally, the abovementioned precursor was preheated at 400 °C for 4 h followed by annealing at 700 °C for 8 h in an Ar/H$_2$ atmosphere (92%/8% in volume ratio) to obtain the NVP/C sample.

## Structural and physicochemical characterization

The crystal structures of the samples were examined with an X-ray diffractometer (X'Pert PRO MPD, Holland) using Cu Kα radiation (40 kV, 40 mA). Raman spectra were obtained with a Jobin-Yvon HR800 instrument. The morphology was characterized by field-emission scanning electron microscopy (SEM, JEOL, JSM-6360LV) and transmission electron microscopy (TEM, Tecnai G12, 200 kV). The cycled Na metal was glued to the sample table in an argon-filled glovebox (H$_2$O and O$_2$ <1 ppm) and transferred to the SEM instrument. Thermogravimetric analysis (TGA) of the COFs was performed from 25 to 800 °C at a heating rate of 10 °C min$^{-1}$ under N$_2$ flow (Pyris Diamond TG/DTA). TGA of the NVP/C sample was conducted from 25 to 800 °C at a heating rate of 10 °C min$^{-1}$ under air flow. X-ray photoelectron spectroscopy (XPS) measurements were carried out with a Perkin-Elmer PHI-5700 ESCA System with monochromated Al Kα radiation. To prevent cycled Na metal from reacting with air, we sealed the samples in tubes in an argon-filled glovebox (H$_2$O and O$_2$ <1 ppm) and transferred them to the XPS instrument. Fourier transform infrared (FTIR) spectroscopy of the powders was performed with a Shimadzu IRTracer-100 using the attenuated total reflection (ATR) infrared mode. The FTIR spectra of the cycled QSSEs in the Na‖Na cells were measured in an argon-filled glovebox (H$_2$O and O$_2$ <1 ppm), and the films were exposed to air for <5 min during sample transfer to the FTIR instrument. UV–vis diffuse reflectance spectra (DRS) were measured with a spectrometer (UV-2700, Shimadzu, Japan) by measuring the reflectance of powders in the solid-state using barium sulfate (BaSO$_4$) as a standard with 100% reflectance. The specific surface area and pore structure were measured by collecting N$_2$ adsorption-desorption isotherms at 77 K (Quantachrome IQ2). The surface area was calculated by the Brunauer–Emmett–Teller method, and the pore size distribution was calculated based on the nonlocal density functional theory (NLDFT, a carbon model containing slit pores) model in Quantachrome ASiQwin software. $^{13}$C and $^{23}$Na solid-state nuclear magnetic resonance (NMR) spectra were collected on a Bruker AMX500 (500 MHz) spectrometer. The zeta potential was determined by using SZ-100-Z (HORIBA, Japan) at 25 ± 1 °C.

## Electrochemical characterization

**Ionic conductivity.** The activated TPBD-NaTFSI and TPDBD-CNa-NaTFSI samples were cold-pressed into pellets at 20 MPa for 5 min and then degassed under vacuum at 120 °C for 12 h to obtain self-standing pellets for the electrochemical tests. The thickness of the electrolyte membrane was ~400 μm. Ionic conductivities were measured in the blocking Ti|QSSE|Ti cell configuration via electrochemical impedance spectroscopy (EIS) with a frequency range from 1 Hz to 1 MHz (overall 85 data points) and an applied amplitude of 100 mV with ~9 wt.% solvents (PC (99.99%, DoDoChem) with 5% FEC (99.9%, DoDoChem)) in an argon-filled glovebox (H$_2$O and O$_2$ <1 ppm). Before the conductivity measurements, the test cells were maintained at each test temperature (from −40 to 100 °C) for at least 60 min to reach equilibrium without additional pressure. At least three cells were tested in a single electrochemical experiment. The ionic conductivities ($\sigma$) were determined according to the following equation:

$$\sigma = \frac{l}{R \times A} \quad (1)$$

where $l$ is the thickness of the pellet (cm), $R$ is the resistance (Ω), and $A$ (cm$^2$) is the area of contact with the electrode. The activation energy (Ea) was determined from the slope of the Arrhenius plot.

**Na-ion transference number.** The Na-ion transference number ($t_{Na+}$) was evaluated using a potentiostatic polarization method at 25 ± 1 °C without additional pressure. A symmetric cell was assembled with two sodium electrodes (300 μm, 99.7%) in an argon-filled glovebox (H$_2$O and O$_2$ <1 ppm) to measure the Na-ion transference number ($t_{Na+}$). At least three cells were tested in a single electrochemical experiment. The DC current flowing through the Na|QSSE|Na symmetric cells and the AC impedance of the cells before and after polarization were measured at 10 steps per decade with a frequency range of 1 MHz to 0.01 Hz under a polarization potential ($\Delta V$) at 100 mV to determine the $t_{Na+}$ value of the TPBD-QSSE and TPDBD-CNa-QSSE according to the following equation:

$$t_{Na+} = \frac{I_{ss}(\triangle V - I_0 R_0)}{I_0(\triangle V - I_{ss} R_{ss})} \quad (2)$$

where $I_{SS}$ is the steady-state current, $I_O$ is the initial current, $\Delta V$ is the applied potential, and $R_O$ and $R_{SS}$ are the interfacial resistances before and after polarization, respectively.

**Linear sweep voltammetry (LSV).** LSV was conducted with Ti|QSSE|Na asymmetric cells in an argon-filled glovebox (H$_2$O and O$_2$ <1 ppm) operated under a sweep rate of 2 mV s$^{-1}$ in a voltage range from 0 to 6.0 V (vs. Na/Na$^+$) at 25 ± 1 °C without additional pressure. At least three cells were tested in a single electrochemical experiment.

**Na plating/stripping experiments.** Na‖Na symmetric cells were assembled with the QSSEs in an argon-filled glovebox (H$_2$O and O$_2$ <1 ppm) and tested under different current densities at 25 ± 1 °C without additional pressure. At least three cells were tested in a single electrochemical experiment.

**Assembly of Na|QSSE|NVP/C cells.** CR2032 coin cells were fabricated by assembling the NVP/C cathode, QSSEs, and a Na anode in an argon-filled glovebox (H$_2$O and O$_2$ <1 ppm). To prepare the positive electrode, NVP/C (0.14 g), QSSEs, ketjenblack (EC-600JD, AkzoNobel), and polyvinylidene difluoride (PVDF, MS-SOLEF-PVDF5130, Shenzhen Kejing Star Technology Company) were homogeneously blended at a mass ratio of 14:3:2:1 in N-methyl-2-pyrrolidone (NMP) and then directly stirred for 3 h (magnetic stirrer, H04-1) in a dry room (25 ± 1 °C, ≤20% humidity level) to obtain a viscous solution. The resulting slurry was uniformly coated on a conductive carbon-coated Al foil (16 μm thickness, Guangdong Zhuguang New Energy Technology Co. Ltd.) and dried in a vacuum oven at 120 °C for 12 h. The NVP/C cathode was cut to a shape 12 mm in diameter for CR2032 assembly. The specific current (mA g$^{-1}$) and specific capacity (mAh g$^{-1}$) mass values reported are based on the mass of active material (1.3–2.6 mg cm$^{-2}$) in the cathode. The masses of the nonelectrochemical active materials were in the range of 0.7–1.4 mg cm$^{-2}$, and that of the current collector was ~4.6 mg cm$^{-2}$. To prepare the QSSEs, ~9 wt.% non-aqueous liquid additive (PC with 5% FEC) was added into TPDBD-CNa-NaTFSI/TPBD-NaTFSI pellets in an argon-filled glovebox (H$_2$O and O$_2$ <1 ppm). The volume of the Na anode was 100 or 300 μm × 0.95 cm$^2$. For pouch cell fabrication, the cathode was cut into a rectangle (3 × 4 cm$^2$ in size), and the total mass was 25.6 mg. Polytetrafluoroethylene (PTFE) was fibrillated to prepare large sheets of TPDBD-CNa membranes[35,41,57]. A mixture of TPDBD-CNa, NaTFSI, and PTFE (polytetrafluoroethylene dispersion) in a weight ratio of 8:1:1 in ethanol was rolled into an ~90 μm freestanding film and then cut into ~5 × 5 cm$^2$ of pieces for the pouch cell. At least three coin and pouch cells were tested in a single electrochemical experiment. All electrochemical tests were carried out

in an environmental chamber (25 ± 1 °C) except when indicated otherwise.

## Computational details

**Theoretical calculation.** Generalized gradient approximation with the Perdew-Burke-Ernzerhof functional (GGA-PBE) was used for the exchange-correlation energy calculations. The commercial version of the DMol3 program[58,59] and spin-polarized calculations were employed with the double numerical polarization basis set. DFT semicore pseudopotential was applied for the core-electron treatment. The Brillouin zone was sampled by a Monkhorst–Pack grid as the Γ-point for all systems with $1 \times 1 \times 1$ k-points. The self-consistent field (SCF) convergence for each electronic energy was set as $1.0 \times 10^{-5}$ Ha, 0.002 Ha Å$^{-1}$ for force and 0.005 Å for displacement.

**Molecular dynamics (MD) model system.** The COF sheets were used to build a simulation diagram. The pore size of the COF was approximately 2.3 nm (diameter). The nanopores were filled with 255 Na$^+$ and TFSI$^-$. The applied electric field along the Z direction was approximately 0.12 V nm$^{-1}$. A significantly higher electric field was applied to obtain more precise statistics for the time scales of the MD calculations. The simulation box had dimensions of 90 Å × 68 Å × 55 Å. Periodic boundary conditions were used in all three directions. In the simulations, all these sheets were fixed.

**MD simulations.** All MD simulations were carried out with the LAMMPS package. VMD software and MATLAB were used to visualize the results. During the simulation, an all-atom OPLS-AA force field was used for the COF. The forced field parameters for NaTFSI were obtained from the literature[40,60]. The cut-off distance for all the short-range van der Waals interactions was set as 12 Å. The long-range electrostatic interaction was computed by using the particle–particle particle–mesh (PPPM) algorithm. All of the data reported in this work were averaged over a set of 5 simulations from various initial velocity distributions. For each system, the simulation was conducted in a canonical ensemble (NVT) at 298 K controlled by the Nose–Hoover thermostat method. The time step was set as 1 fs, and the data were collected every 1 ps. The total simulation time of each model was 12 ns.

## Data availability

Data supporting the findings of this work are available within the article and its Supplementary Information file. Source data are provided with this paper.

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

## Acknowledgements

The work was supported by the National Natural Science Foundation of China (Nos. 51972342, 52272259, 52202311, and 52073305), Shandong Provincial Natural Science Foundation, China (ZR2020QE050), Taishan Scholar Project of Shandong Province (ts20190922), Key Basic Research Projects of Natural Science Foundation of Shandong province (ZR2019ZD51), Project funded by China Postdoctoral Science Foundation (2022M722684).

## Author contributions

Z.J.F., Z.L., and Y.C.Y. supervised and designed the experiments. Y.C.Y. prepared the electrode materials and performed the electrochemical battery characterization. Y.C.Y. wrote the draft manuscript. Z.J.F., Z.L., and Y.C.Y. revised the manuscript. B.Q. conducted SEM characterization. Y.C.Y. carried out the DFT calculation and analysis. T.W. and Y.G.Y. performed the MD simulation and analysis. T.W., W.N.L., Z.P.Q., C.L.C., C.HF., and G.W.W. helped analyze the experimental and characterization data. All authors contributed to the discussion and preparation of this paper.

## Competing interests

The authors declare no competing interests.
