## [Peer Review File · Nature Communications]

REVIEWER COMMENTS

Reviewer #1 (Remarks to the Author):

This manuscript reports a –COO– modified covalent organic framework (COF) as Na⁺-conducting quasi-solid-state electrolyte (QSSE) for Na metal batteries. It is of significance to come up with solid or quasi-solid-state electrolytes with high conductivity and wide electrochemical windows for advanced Na batteries, and the work deals with this important topic. Through a series of experimental, DFT, and MD analyses, the authors demonstrated that the carboxylic acid groups (–COO–) anchored on the COF inwalls construct sub-nanochannels that can dissociate NaTFSI and thus transport Na ions while enhancing the oxidative stability of the polymer electrolytes. The proposed electrolyte exhibited high ionic conductivity and Na⁺ transference number and facilitated uniform Na plating/stripping. The paper would be interesting to researchers active in the area of solid-state Na batteries. However, the manuscript needs improvement as commented below:

1. Need more detailed explanations on the "Pendulum" transport mechanism based on the constituents and structures of QSSE.
2. The reviewer believes that a key finding is the high oxidative stability of QSSE of ~5.3 V; however, the mechanism behind the improved stability is not clearly presented. The DFT results are only given without detailed answers on why and how the combinations of various constituents induce the extension of the window.
3. The composition of the composite electrode should be clearly described in the Experimental section. For practical applications, the proposed electrolyte should be able to provide an ionic conducting pathway in the composite cathode. In full cell demonstrations, however, the mass loading of the cathode (1.3 mg/cm²) is too low to prove the efficacy of QSSE.
4. The authors highlighted the high conductivity at low temperatures; however, no data are presented to demonstrate Na plating-stripping and/or full cell operations using QSSE at reduced temperatures.

5. In Fig. 7(e), the voltage profiles of the two cells appear different in shape. In principle, the polarizations can be affected by the electrolyte properties, but the overall shape of the voltage profile should be similar when the same electrode materials are used.

Reviewer #2 (Remarks to the Author):

The manuscript demonstrated a quasi-solid electrolyte for sodium batteries by using a –COO– modified covalent organic framework. The structures of COF and COF based electrolyte are basically characterized, and new concept of Na⁺ and TFSI⁻ distribution is proposed with theoretical calculations. However, similar concept of using functionalized COFs as electrolyte in Li batteries has been reported, and the performance was not as excellent as expected. Therefore, I suggest a major revision and then reconsideration in high-impact Nature Communications journal, and the comments are listed below.

(1) The electrochemical performance of the symmetric battery is at an average level, especially under such small current density of 0.01 mA cm⁻². (Nat. Mater. 2022, 21, 1057, 0.5 mA cm⁻²; Nano-Micro Lett. 2021, 13, 105, 0.4 mA cm⁻²; Adv. Mater. 2021, 33, 2100353, 0.1 mA cm⁻²; Adv. Mater. 2021, 33, 2105329, 1 mA cm⁻²).

(2) The XRD pattern of TPDBD should be provided to shown the crystallinity, and it is obvious that the pore structure in Fig. S4 does not match well with the theoretical ones, such as surface area, pore size. The surface area and pore size of COFs can be theoretically predicted, while the N₂ adsorption isothermal shows weak porosity. For pore size distribution, the NLDFT or QSDFT, and the pore geometry and other parameters should be given, otherwise it is misleading or far from actual ones.

(3) In my viewpoint, the COFs, especially with -COOHs and COONa are amorphous with weak crystallinity, according to the BET, XRD etc. So how the aligned 1D nanochannels can be used for Na ion transport. And the concept of preferentially adsorb Na⁺ on the COF inwalls with TFSI⁻ repulsed at the center of COF channels, and PC in the sub-nanoscale channels, still need more evidence.

(4) Fig. S18 is hard to understand to distinguish the CIP and AGG, since the peaks are so broad.

(5) How about the performances of symmetrical cells and full cells, compared with the previously reported values? Such as providing a table for compare.

(6) The cycling performances of the Na₃V₂(PO₄)₃/C|QSSEs|Na full battery (coin cells) with a low cathode mass loading (1.3 mg cm⁻²) and a thick Na anode (300 μm) under 0.5 C is not enough for assessing its practical application. Commonly, the lower N/P ratio or anode-free coin full cells is adopted to confirm the practical application of the quasi-solid electrolyte in sodium batteries. Unfortunately, it cannot be found in this manuscript.

(7) Moreover, pouch cells should be used in demonstrating the commercialization prospects of the quasi-solid electrolyte. Unfortunately, it also cannot be found in this manuscript. The evaluation is suggested to be provided.

(8) The thickness of the membrane is too thick (400 μm), which adversely affects the energy density of the batteries. The decrease in thickness should be considered.

(9) How about the performances without adding any PC solvent, and in this case the intrinsically ionic conductivity can be evaluated?

(10) For the specific surface area of TPDBD-CNa-NaTFSI and TPBD-NaTFSI are 38 and 28 $\text{m}^2 \text{g}^{-1}$, respectively. And the weight loss of TPDBD-CNa and TPDBD are shown as 44% and 46%, respectively. From the curves of Fig. S14, this could be due to measurement error.

(11) For studying the chemical composition of the solid electrolyte interface (SEI), this manuscript only carried out the XPS experiments of surface. But it is far from enough for the SEI analysis. For example, the XPS depth profiling analysis or the time-of-flight secondary ion mass spectrometry depth profiling analysis should be considered.

(12) For Table S2, the thickness of electrolyte should be also included for better comparison.

(13) Other minor issues should be taken seriously:

a) The abbreviation of Fourier transform infrared spectroscopy should be unify, such as FT-IR in line 215 while FTIR in 162.

b) In line 192, the “a” in “-NaTFSI” should be in “Times New Roman” style.

c) The format of references should be uniform, such as title capitalization and journal abbreviations.

Reviewer #3 (Remarks to the Author):

In this article, authors fabricated quasi-solid polymer electrolyte using TPBD-NaTFSI and TPDBD-CNa-NaTFSI as the support substrate by absorbing PC and FEC solvent. In this way, high ion conductivity and high oxidative potential is achieved.

The strength of this work is the biomimetic inspired synthesis of a new type of COF materials as the host for Na ions conduction.

However, the weakness is also not insignificant. Specifically, because of the existence of solvent, what determines the Na ions transport is the solution, unlike the case of ion conduction via polymer chains in

the polymer electrolytes. Authors repeatedly emphasizes that the -COO- functional group anchored on the COF wall promotes the decomposition of NaTFSI and boosts the Na ion transport. The calculation was conducted based on this assumption. However, this assumption is questionable. Another more possible reason to the enhanced ion conductivity is that the functionalized COF has a stronger capability in retaining the solvent solution. The observed effect is not limited to such COF. Instead, similar effect could also be realized from materials that have a high specific surface area, porous structure and solvent-capturing functional groups, for example, molecular sieves and OH- functionalized cellulose. The conclusion will not be convincing unless authors can prove that the Na ions conduct in the COF host much faster than in solution, and/or the effect of PC solvent to Na ion conduction has been explicitly clarified.

Other minor issues:

1) it is not meaningful to test the thermal stability of pure TPBD-NaTFSI and TPDBD-CNa-NaTFSI; instead, it is more relevant for battery practice to test the thermal stability of their mixture with solvent.

2) the LSV test shows a wide electrochemical stability window of 5.32 V. This is good for the electrolyte membrane itself but it does not mean the full-cell battery can deliver such high voltage. The actual output voltage as well as power performance of a Na-ion battery based on quasi-solid polymer electrolytes is more significantly affected by the interface property. So it could be misleading to claim a high-performance full cell based on high-performance of the electrolyte membrane.

Response to Reviewers' Comments

Response to Reviewer #1's Comments:

This manuscript reports a $-\text{COO}^-$ modified covalent organic framework (COF) as Na^+ -conducting quasi-solid-state electrolyte (QSSE) for Na metal batteries. It is of significance to come up with solid or quasi-solid-state electrolytes with high conductivity and wide electrochemical windows for advanced Na batteries, and the work deals with this important topic. Through a series of experimental, DFT, and MD analyses, the authors demonstrated that the carboxylic acid groups ($-\text{COO}^-$) anchored on the COF inwalls construct sub-nanochannels that can dissociate NaTFSI and thus transport Na ions while enhancing the oxidative stability of the polymer electrolytes. The proposed electrolyte exhibited high ionic conductivity and Na^+ transference number and facilitated uniform Na plating/stripping. The paper would be interesting to researchers active in the area of solid-state Na batteries. However, the manuscript needs improvement as commented below:

Reply: Thanks for your thoughtful comments. We have tried our best to revise the manuscript according to your valuable comments and suggestions.

1. Need more detailed explanations on the "Pendulum" transport mechanism based on the constituents and structures of QSSE.

Reply: Many thanks for the reviewer's valuable comment and suggestion. According to the reviewer's suggestion, we provide a more detailed description of the "Pendulum" transport mechanism in the revised manuscript (Please see Highlights in Page 11 of the revised manuscript). The relevant discussion is as follows: due to the electronegative modification of $-\text{COO}^-$ groups, Na^+ is preferentially adsorbed in the sub-nanometer-sized zones (constructed by the adjacent $-\text{COO}^-$ groups and COFs inwalls), while TFSI⁻ is repulsed at the center of COF channels. Such sub-nanometer-sized zones are like the biomimetic Na^+ transfer channels, which can effectively promote the Na^+ transport efficiency directionally along the oxygen atoms. In fact, the jumping transfer mode of Na^+ from one carboxylic/carbonyl position to the next unoccupied site under a given voltage can be regarded as a kind of "Pendulum" Na^+ transport mechanism, yielding a high Na^+ conduction performance.

2. The reviewer believes that a key finding is the high oxidative stability of QSSE of ~ 5.3 V; however, the mechanism behind the improved stability is not clearly presented. The DFT results are only given without detailed answers on why and how the combinations of various constituents induce the extension of the window.

Reply: Thanks for your valuable comment. According to the reviewer's request, we provide a detailed analysis of DFT results to illustrate the mechanism of high stability of QSSE. Moreover, we have also performed additional Fourier transform infrared spectra (FTIR) and thermogravimetric analysis (TGA) experimental characterizations to confirm the high stability of QSSE under various constituent conditions. To reveal the high oxidative stability of QSSEs at molecular level, the optimized structures and

45 highest-occupied/lowest-unoccupied molecular orbitals (HOMO or LUMO) of TPBD
and TPDBD were calculated by DFT calculations (**Figure R1**). It should be noted that
the lower HOMO energy reflects the harder ability to lose electrons, corresponding to
a higher oxidation potential. Due to the introduction of electron-withdrawing $-\text{COOH}$
groups, the TPDBD shows lower HOMO energy (-5.31 eV) than TPBD (-4.83 eV),
indicating stronger antioxidant property and higher decomposition potential of TPDBD
(*Angew. Chem. Int. Ed.* **60**, 24915 (2021), *Chem. Mater.* **31**, 4598 (2019)). Moreover,
the band gap (E_g) of TPDBD is determined to be 2.17 eV, larger than that of TPBD
(1.90 eV), further implying the better electron-insulating property of TPDBD, which is
more reliable for the solid-state electrolyte. FTIR spectrum of TPDBD-CNa-QSSE
shows stronger propylene carbonate (PC)- Na^+ interactions (720 and 1403 cm^{-1}) and
concentrated TFSI $^-$ (787 cm^{-1}) than TPBD-QSSE, demonstrating stronger electrolyte
aggregation effect (**Figure R2a**). Due to the high confinement of TPDBD-CNa, a tiny
amount of PC tends to be rapidly captured into the sub-nanometer channels of TPDBD-
CNa-QSSE. As shown in **Figure R2b**, the TPDBD-CNa-QSSE and TPBD-QSSE are
more thermostable than pure PC and glass fiber separator filled with PC, making it
harder to oxidize and decompose the quasi-solid-state electrolyte, further verifying the
excellent interfacial contact and compatibility of PC confined in the QSSEs. Therefore,
the oxidative decomposition of the quasi-solid-state electrolyte requires a higher force,
which is another reason for the high voltage window of TPDBD-CNa-QSSE (*Nat.*
*Commun.* **13**, 1510 (2022)). The experimental and theoretical results indicate a high
stability (including high oxidative stability) of both COFs and QSSEs.

In this revision, we have supplemented this **Figure R2b** into the Supplementary
Information as **Supplementary Figure 26**, and also added corresponding discussions
to both the context to clarify this issue in the manuscript. (Please see Highlights in Page
10, and 11 of the revised manuscript).

**Figure R1.** Calculated HOMO and LUMO energy of TPBD and TPDBD based on the
DFT calculations.

Figure R2. (a) FTIR spectra of the prepared TPDBD-CNa-QSSE, TPBD-QSSE, and PC solvent (cycled COF membranes in Na||Na cells after 20 cycles). (b) TGA curves of the COF-based membranes and glass fiber with solvent.

3. The composition of the composite electrode should be clearly described in the Experimental section. For practical applications, the proposed electrolyte should be able to provide an ionic conducting pathway in the composite cathode. In full cell demonstrations, however, the mass loading of the cathode (1.3 mg/cm^2) is too low to prove the efficacy of QSSE.

Reply: Many thanks for the reviewer's instructive suggestion. A detailed description of the composite electrode was added in the revised manuscript (Please see Highlights in Page 21 and 22 of the revised manuscript). CR2032 coin batteries were fabricated by assembling NVP/C cathode, QSSE, and Na anode in an argon-filled glove box (H_2O and $\text{O}_2 < 1 \text{ pmm}$). The cathodes were prepared by homogeneously blending NVP/C (0.14 g), QSSEs, acetylene black, and polyvinylidene difluoride (PVDF) at a mass ratio of 14:3:2:1 in N-Methyl-2-pyrrolidone (NMP), and then directly stirring for 3 h to get a viscous solution. The resulting slurry was uniformly coated on a conductive carbon-coated Al foil and dried in a vacuum oven at $120 \text{ }^\circ\text{C}$ for 12 h. The NVP/C cathode was cut into 12 mm in diameter for CR2032 assembly. The mass of specific current (mA g^{-1}) refers to the mass of active material ($1.3\text{-}2.6 \text{ mg cm}^{-2}$) in the cathode, and the non-electrochemical active materials was $0.7\text{-}1.4 \text{ mg cm}^{-2}$, and current collector was about 4.6 mg cm^{-2} . The volume of Na anode was 100 or $300 \text{ } \mu\text{m} \times 0.95 \text{ cm}^2$. As for pouch cell fabrications, the cathode was cut into a rectangle ($3 \times 4 \text{ cm}^2$ in size), and the total mass is 25.6 mg . All the electrochemical tests were carried out at $25 \pm 1 \text{ }^\circ\text{C}$ except the especial declaration.

Also, we had investigated the electrochemical performances of TPDBD-CNa-QSSE battery with high mass loading of active materials in the cathode (2.6 mg cm^{-2}) and thin Na anode ($\sim 100 \text{ } \mu\text{m}$). According to the editor's suggestion, the specific current (mA g^{-1}) values instead C-rate values when electrochemical energy storage tests are discussed. 1 C charging and discharging rate here was defined as 120 mA g^{-1} . As shown in **Figure R3**, the NVP/C|TPDBD-CNa-QSSE|Na retains 63 mAh g^{-1} after 100 cycles at 60 mA g^{-1} with a coulombic efficiency of 99.6%, and displays 99.7 mAh g^{-1} after 68 cycles at 12 mA g^{-1} with a coulombic efficiency of 99.6%. When the current densities are 12, 24,

60, and 120 mA g⁻¹, the specific capacities of NVP/C|TPDBD-CNa-QSSE|Na are 101.5,
106, 91.6 and 78.5 mAh g⁻¹, respectively. Even at a higher rate of 240 mA g⁻¹, the
battery still represents 54.5 mAh g⁻¹. When the current density recovers to 60 mA g⁻¹,
a specific capacity of 103.9 mAh g⁻¹ can be obtained, revealing reversible
characteristics. Those results verify that the NVP/C|TPDBD-CNa-QSSE|Na full cell
with high mass loading of cathode shows acceptable rate performance and cycle
stability.

In this revision, we have supplemented this **Figure R3** into the Supplementary
Information as **Supplementary Figure 46**, and also added corresponding discussions
to both the context and Supplementary Information to clarify this issue (Please see
Highlights in Page 16 of the revised manuscript and Page 34 of Supporting Information).

**Figure R3.** (a) Photo of Na anode thickness. (b) Cycle stability at 60 mA g⁻¹, (c) cycle
stability at 12 mA g⁻¹, and (d) rate performance of the NVP/C|TPDBD-CNa-QSSE|Na
with high mass loading of active materials (2.6 mg cm⁻²) in the cathode.

*4. The authors highlighted the high conductivity at low temperatures; however, no data*
*are presented to demonstrate Na plating-stripping and/or full cell operations using*
*QSSE at reduced temperatures.*

**Reply:** Many thanks for your valuable comment. According to the reviewer's
suggestion, we performed the Na plating-stripping at 0, -5, -10, -15, -20, and -25 °C
and full cell operations at 0 °C. As shown in **Figure R4a**, as the temperature decreased
from 0 to -20 °C, Na|TPDBD-CNa-QSSE|Na represents low overpotential and stable
voltage profiles under 0.01 mA cm⁻². When the temperature further decreased to the
severe -25 °C, irregular voltage fluctuations emerge, which can be ascribed to the
unstable Na plating/stripping and formation of dead Na (*Nano Lett.* **19**, 8664 (2019),
*ACS Energy Lett.* **5**, 2411 (2020)). However, due to the absence of biomimetic sub-
nanochannels, the voltage irregularity of Na|TPBD-QSSE|Na symmetrical cell appears

at relative higher temperature of $-10\text{ }^{\circ}\text{C}$. Figure R4b exhibits relatively stable voltage
 curves for over 550 h without obvious overpotential fluctuation at $0\text{ }^{\circ}\text{C}$, further
 indicating that the bioinspired design of ionic channels is beneficial for the uniform
 deposition/stripping of Na^+ . In addition, the NVP/C|TPDBD-CNa-QSSE|Na cell (active
 material mass loading of 2.0 mg cm^{-2}) at $0\text{ }^{\circ}\text{C}$ shows 101.2 mAh g^{-1} reserved after 75
 cycles at 12 mA g^{-1} with a coulombic efficiency of 99.4% (Figure R4c), confirming
 excellent cycling stability of QSSE based full cell at decreased temperatures.

In this revision, we have supplemented this Figure R4a-b and Figure R4c into the
 Supplementary Information as Supplementary Figure 35 and Supplementary Figure
 47, respectively, and also added corresponding discussions to both the context to clarify
 this issue in the manuscript (Please see Highlights in Page 12 and 17 of the revised
 manuscript).

**Figure R4.** (a) Temperature-dependent galvanostatic cycling of Na stripping-plating
 test of Na|QSSEs|Na at 0, -5 , -10 , -15 , -20 , and $-25\text{ }^{\circ}\text{C}$ with a current density of 0.01
 152 mA cm^{-2} . (b) Long cycle performance of Na|TPDBD-CNa-QSSE|Na at $0\text{ }^{\circ}\text{C}$ with a
 153 current density of 0.01 mA cm^{-2} . (c) Cycle performance of NVP|TPDBD-CNa-
 154 QSSE|Na at 12 mA g^{-1} at $0\text{ }^{\circ}\text{C}$.

 5. In Fig. 7(e), the voltage profiles of the two cells appear different in shape. In principle,
 the polarizations can be affected by the electrolyte properties, but the overall shape of
 the voltage profile should be similar when the same electrode materials are used.

**Reply:** Thanks for your important comment. To analyze the charging and discharging
 behavior of the QSSE-based cell scrupulously, we carried out the differential capacity
 (dQ/dV) profiles based on the voltage profiles. At a current density of 12 mA g^{-1} , the
 NVP/C|TPDBD-CNa-QSSE|Na mainly presents the charging/discharging platforms at
 $3.40/3.30\text{ V}$ (Figure R5a), corresponding to the reversible transformation of $\text{V}^{3+}/\text{V}^{4+}$,

exhibiting faster Na^+ reaction kinetic of NVP/C|TPDBD-CNa-QSSE|Na during Na
 insertion/extraction process. As the current density increased to 120 mA g^{-1} , two pairs
 of well-defined redox peaks appear in dQ/dV profiles, which can be attributed to the
 dynamic hysteresis (**Figure R5c**). However, due to the absence of biomimetic sub-
 nanochannels, the dQ/dV profile of NVP/C|TPBD-QSSE|Na exhibits multiple
 platforms even at a small current density of 12 mA g^{-1} (**Figure R5d**), manifesting
 serious kinetic hysteresis phenomenon. The above results verify that the
 electrochemical behavior of NVP/C cathode in different cells is intrinsically consistent,
 and the different voltage profile shapes are mainly due to the Na^+ transport efficiency
 and kinetic hysteresis reaction.

 **Figure R5.** (a) Differential capacity (dQ/dV) profiles of NVP/C|TPDBD-CNa-
 QSSE|Na at 12 mA g^{-1} . (b) dQ/dV profiles of NVP/C|TPDBD-CNa-QSSE|Na at 60 mA g^{-1} .
 (c) dQ/dV profiles of NVP/C|TPDBD-CNa-QSSE|Na at 120 mA g^{-1} . (d) dQ/dV
 profiles of NVP/C|TPBD-QSSE|Na at 12 mA g^{-1} .

**Response to Reviewer #2's Comments:**

*The manuscript demonstrated a quasi-solid electrolyte for sodium batteries by using a*
*–COO– modified covalent organic framework. The structures of COF and COF based*
*electrolyte are basically characterized, and new concept of Na⁺ and TFSI⁻ distribution*
*is proposed with theoretical calculations. However, similar concept of using*
*functionalized COFs as electrolyte in Li batteries has been reported, and the*
*performance was not as excellent as expected. Therefore, I suggest a major revision*
*and then reconsideration in high-impact Nature Communications journal, and the*
*comments are listed below.*

**Reply:** Thanks for your constructive comments. We have tried our best to revise the
manuscript, and the valuable comments have been elaborately solved in the revision,
and please see more details below.

*(1) The electrochemical performance of the symmetric battery is at an average level,*
*especially under such small current density of 0.01 mA cm⁻². (Nat. Mater. 2022, 21,*
*1057, 0.5 mA cm⁻²; Nano-Micro Lett. 2021, 13, 105, 0.4 mA cm⁻²; Adv. Mater. 2021,*
*33, 2100353, 0.1 mA cm⁻²; Adv. Mater. 2021, 33, 2105329, 1 mA cm⁻²).*

**Reply:** Thank you for the valuable comments. Considering that the test conditions have
a great influence on the final electrochemical performance of electrode, we provide a
comparison of experimental details in the recommended references as shown in **Table**
**R1**. Actually, the EO10-PFPE/Solupor composite electrolyte was used with the
assistance of commercial Solupor separator, and the symmetrical Na cell was performed
at 80 °C (*Nat. Mater.* 2022, 21, 1057). PLM@LE was compounded with a large amount
of liquid electrolyte (90 μL NaClO₄-PC with 5% FEC, *Nano-Micro Lett.* 2021, 13, 105).
m-PPL was prepared by poly(methyl methacrylate)–polystyrene (PMMA–PS) interface
layers closely attached on both sides of the PE separator, and a symmetrical Li cell was
tested at 60 °C (*Adv. Mater.* 2021, 33, 2100353). UFF/PEO/PAN/LiTFSI SSE
symmetrical Li cell was performed at 50 °C (*Adv. Mater.* 2021, 33, 2105329). Therefore,
there is no unified standard for the preparation of solid-state electrolyte films and the
electrochemical test conditions are also non-uniform.

By investigating the experimental details, we find that the conventional current
density of COFs-based QSSEs in the symmetric battery was about 0.01-0.05 mA cm⁻².
Moreover, the COFs membrane in our work was fabricated under 20 MPa for 5 min,
much lower than the reported COFs-based QSSEs (> 200 MPa, 40 min) (**Table R1**).
For comparison, we investigated the tripping-plating plots of Na|TPDBD-CNa-
QSSE|Na cell at high current densities of 0.02, 0.05, and 0.1 mA cm⁻² without
additional pressure. As shown in **Figure R6**, the TPDBD-CNa-QSSE symmetrical cell
shows lower polarization and stable Na plating/stripping performance without any sign
of short circuits. Moreover, the symmetric battery of TPDBD-CNa-QSSE at 0.05 mA
221 cm⁻² exhibits steady Na insertion/extraction processes for over 450 h without obvious
fluctuation of potential, indicating that there is good interfacial stability between QSSE
and Na metal anode. More significantly, the Na⁺ conduction mechanism is deeply
analyzed through the biomimetic concept for the first time in our work, and ample

evidence has been carried out using the sodium ion full and soft pack battery to assess
 the practical application of our QSSE, which may provide profound implications for
 the promising energy storage fields.

In this revision, we have supplemented this **Figure R6** into the Supplementary
 Information as **Supplementary Figure 34**, and also added corresponding discussions
 to both the context to clarify this issue in the manuscript (Please see Highlights in Page
 12 of the revised manuscript).

**Figure R6.** (a) Na stripping-plating test of Na|TPDBD-CNa-QSSE|Na at current
 density of 0.02 and 0.1 mA cm⁻² for 2 h per cycle. (b) Na stripping-plating test of
 Na|TPDBD-CNa-QSSE|Na at current density of 0.05 mA cm⁻² for 450 h and 2 h per
 cycle.

**Table R1.** Comparison of typical Na plating/stripping performance for the reported
 literature.

Materials	Li ⁺ /Na ⁺	Additional components	Current densities (mA cm ⁻²)	Time (h)	Thickness (μm)	Method of making film	Ref.
EO10-PFPE/ Solupor composite electrolyte	Na ⁺	NA	0.5 (80 °C)	1000	80-90	perfluoropolyether (PFPE) domains into PEO-based electrolytes	1
PLM@LE	Na ⁺	90 μL (NaClO ₄ -PC with 5% FEC)	0.6	~300	120	PLM: polytetrafluoroethylene (PTFE)=9:1 by hand grinding (PMMA-PS)	2
m-PPL	Li ⁺	NA	0.1 (60 °C)	1500	10	closely attached on both sides of the PE separator	3
UFF/PEO/PA N/LiTFSI SSE	Li ⁺	9.6% (DMF and negligible acetonitrile)	1 (50 °C)	~1140	4.2	PAN/LiTFSI/DMF solution was tape casted on the UFF/PEO/LiTFSI film	4

TpPa-SO ₃ Li	Li ⁺	NA	0.01	320	50	Cold-pressing (200 kN)	5
Tp-PaSO ₃ Li-COF	Li ⁺	10 μL EC/DMC (v/v = 1:1)	NA	NA	100	Cold-pressing (280 Mpa, 5 min)	6
LiCON-3	Li ⁺	NA	0.1	~450	200	Cold-pressing (200 Mpa, 40 min)	7
NaOOC-COF	Na ⁺	10 mL, 1.0 M NaPF ₆ /PC	0.05	700	200	Cold-pressing (200 MPa, 40 min)	8
MIL-121	Li ⁺ /Na ⁺	PC	NA	NA	1000	Cold-pressing	9
LiOOC - COF ₃	Li ⁺	10 μL LiPF ₆ , EC/DEC v/v=1:1)	0.05	320	200	Cold-pressing (250 MPa, 40 min)	10
TPDBD-CNa-QSSE	Na ⁺	~9 wt. % solvent	0.01	1000	400	Cold-pressing (20 MPa, 5 min)	This work
			0.05	450	400		

*Unless otherwise noted, tests were conducted at 25 °C.

References

- Wang, X. et al. Ultra-stable all-solid-state sodium metal batteries enabled by
perfluoropolyether-based electrolytes. *Nat. Mater.* **21**, 1057-1065 (2022).
- Zhang, G. et al. Pancake-like MOF solid-state electrolytes with fast ion
migration for high-performance sodium battery. *Nano-Micro Lett.* **13**, 105
(2021).
- Wang, Z., Shen, L., Deng, S., Cui, P. & Yao, X. 10 μm-thick high-strength
solid polymer electrolytes with excellent interface compatibility for flexible all-
solid-state lithium-metal batteries. *Adv. Mater.* **33**, 2100353 (2021).
- He, F., Tang, W., Zhang, X., Deng, L. & Luo, J. High energy density solid state
lithium metal batteries enabled by sub-5 microm solid polymer electrolytes. *Adv.*
*Mater.* **33**, 2105329 (2021).
- Jeong, K. et al. Solvent-free, single lithium-ion conducting covalent organic
frameworks. *J. Am. Chem. Soc.* **141**, 5880-5885 (2019).
- Li, J. et al. A pre-synthetic strategy to construct single ion conductive covalent
organic frameworks. *Chem. Commun.* **56**, 2747-2750 (2020).
- Li, X. et al. Solution-processable covalent organic framework electrolytes for
all-solid-state Li-organic batteries. *ACS Energy Lett.* **5**, 3498-3506 (2020).
- Zhao, G. et al. COFs-based electrolyte accelerates the Na⁺ diffusion and
restrains dendrite growth in quasi-solid-state organic batteries. *Nano Energy* **92**,
106756 (2022).
- Zettl, R. et al. High Li⁺ and Na⁺ conductivity in new hybrid solid electrolytes
based on the porous MIL-121 metal organic framework. *Adv. Energy Mater.* **11**,
2003542 (2021).
- Zhao, G. et al. COF-based single Li⁺ solid electrolyte accelerates the ion
diffusion and restrains dendrite growth in quasi-solid-state organic batteries.
*Carbon Energy*, 1-13 (2022).

(2) The XRD pattern of TPDBD should be provided to show the crystallinity, and it is
obvious that the pore structure in Fig. S4 does not match well with the theoretical ones,
such as surface area, pore size. The surface area and pore size of COFs can be
theoretically predicted, while the N₂ adsorption isothermal shows weak porosity. For
pore size distribution, the NLDFT or QSDFT, and the pore geometry and other
parameters should be given, otherwise it is misleading or far from actual ones.

**Reply:** Thanks very much for your insightful question. According to the reviewer's
suggestion, we provide the XRD pattern of TPDBD in the Supplementary Information
as **Supplementary Figure 2**. Firstly, the XRD pattern of TPBD exhibits excellent
crystallinity with a specific 2 θ peak at a low angle of 3.4°, belonging to the (100) plane.
After modification of the –COOH groups, the XRD pattern of TPDBD shows a wider
interface distribution at low angles, indicating a weaker crystallinity compared to TPBD
(**Figure R7a**). Such weak crystallinity of TPDBD can be attributed to that the presence
of –COOH groups can disturb the π - π stacking interactions among the layers in the
extended TPDBD structure and bring larger steric hindrance and random displacement
in the 2D layers (**Figure R7b-c**) (*J. Am. Chem. Soc.* **138**, 2823 (2016), *J. Am. Chem.*
*Soc.* **140**, 896 (2018)).

Moreover, we have retested and analyzed the surface area and pore size, and the
relevant discussion has also been revised in the manuscript. Upon repeated tests and
careful analysis, the Brunauer-Emmett-Teller (BET) surface area and total pore volume
were calculated from the N₂ sorption isotherms at 77 K. As shown in **Figure R8**, the
surface area of each COF was calculated using the BET model with relatively good
fitting. The pore size distribution was calculated based on the N₂ sorption isotherm by
using the nonlocal density functional theory (NLDFT, a carbon model containing slit
pore) model in the Quantachrome ASiQwin software. The BET surface area and total
pore volume of TPBD are 281 m² g⁻¹ and 0.42 cm³ g⁻¹, respectively, which decreases
to 74 m² g⁻¹ and 0.30 cm³ g⁻¹ of TPDBD due to the internally cross-linked protic acid.
After Na-activation, the specific surface and pore volume of TPDBD-CNa re-increases
to 116 m² g⁻¹ and 0.60 cm³ g⁻¹ by eliminating the protic acid cross-linked oligomers
inside the pores and layers. Compared with TPBD, the BET surface area and pore
volume of TPDBD exhibit obvious decrease due to the weak crystallinity induced by
the –COO⁻ modification, that is consistent with the PXRD study (**Figure R7a**).

In this revision, we have supplemented this **Figure R8** into the Supplementary
Information as **Supplementary Figure 4**, and also added corresponding discussions
the context to clarify this issue in the manuscript (Please see Highlights in Page 5 and
20 of the revised manuscript).

Figure R7. (a) PXRD patterns for TPBD, TPDBD, and TPDBD-CNa. Optimized structure of (b) TPDBD and (c) TPBD.

Figure R8. N₂ adsorption and desorption curves, pore size distributions, and good fitting plots based on the BET model of TPBD (a), TPDBD (b), and TPDBD-CNa (c).

(3) In my viewpoint, the COFs, especially with -COOHs and COONa are amorphous
with weak crystallinity, according to the BET, XRD etc. So how the aligned 1D
nanochannels can be used for Na ion transport. And the concept of preferentially
adsorb Na⁺ on the COF inwalls with TFSI⁻ repulsed at the center of COF channels, and
PC in the sub-nanoscale channels, still need more evidence.

**Reply:** Thanks for your valuable comments. In order to further verify the concept of
biomimetic design of COFs, we carried out more experimental and theoretical
investigations. As verified in the manuscript, FTIR, ¹³C cross-polarization magic-angle-
spinning (CP-MAS) solid-state nuclear magnetic resonance (NMR), and X-ray
photoelectron spectroscopy (XPS) spectra of TPDBD have successfully proved the
construction of COF skeletons. Like comment 2, after the introduction of -COOH or
COONa, the crystallinity of COFs becomes weak due to the larger steric hindrance and
weak π - π stacking interactions (**Figure R7**). More importantly, the COFs with partial
weak crystallinity can effectively eliminate the grain boundaries, which is conducive to
fast ion transport (*J. Am. Chem. Soc.* **140**, 896 (2018), *ACS Energy Lett.* **5**, 3498 (2020),
*Adv. Mater.* **33**, 2102634 (2021), *Adv. Energy Mater.* **12**, 2200057 (2022)).

Besides, compared with TPBD, the pore size of TPDBD is slightly larger due to the
weak crystallinity induced by the -COO⁻ modification (**Figure R8**). After filling
NaTFSI, TPDBD exhibits an almost identical pore size distribution to TPBD (**Figure**
**R9**). In consideration of the high crystallinity of TPBD, we believe that the aligned sub-
nanochannels of TPDBD can be well reserved due to the similar pore size distributions
of TPBD and TPDBD, and the limited crystallinity is only related to the disruption of
weak π - π stacking interactions. Therefore, in the follow-up simulation analysis, we
adopted the 1D channel model construction to analyze the ion conduction mechanism.
On the premise of maintaining well-aligned sub-nanochannels, the concept of
“preferentially adsorb Na⁺ on the COF inwalls with TFSI⁻ repulsed at the center of COF
channels” of the Na⁺ transfer mechanism was detailly verified by combining
experimental and theoretical analysis. Based on the strong electronegative property of
-COOH groups anchored on COFs, the electropositive Na⁺ is preferentially adsorbed
on such inwalls and the TFSI⁻ is electrostatically repulsed at the center of channels,
which has been confirmed by the zeta potential values, DFT calculations, MD
simulations, and XPS spectra (**Figure R10-R11**).

According to the reviewer’s comment, we further investigate the effect of PC solvent
in the sub-nanoscale channels of COFs using FTIR, DFT, and MD simulations. Due to
the strong wettability, the tiny addition of PC solvent can be rapidly adsorbed into COF
membrane through thermodynamic motion. MD simulation shows that the PC
molecules are uniformly adsorbed and infiltrated on the TPDBD-CNa skeletons while
aggregated on TPBD surface, indicating that the biomimetic design is beneficial for the
uniform wettability of PC on the grain boundaries of COFs (**Figure R12**). As shown in
**Figure R13a**, TPDBD-CNa-QSSE behaves stronger propylene carbonate (PC)-Na⁺
interactions (720 and 1403 cm⁻¹) and concentrated TFSI⁻ (787 cm⁻¹) than TPBD-QSSE,
demonstrating stronger electrolyte aggregation effect (*Nat. Commun.* **13**, 1510 (2022)).
The Na-O bond length of NaTFSI in QSSEs becomes significantly longer, indicating
PC can effectively dissociate Na⁺ and TFSI⁻ (**Figure R13b**). The TGA on COF-based

membranes with solvent was performed (**Figure R13c**). Due to the sub-nanometer
confinement effects promoted by the COFs, the solvent tended to be attached to the
interior of COFs and decomposed more slowly than pure solvent and glass fiber/solvent,
making it harder to oxidize and decompose the quasi-solid-state electrolyte, and further
verifying the excellent interfacial contact and compatibility of PC confined in the
QSSEs, thus effectively boosting the Na^+ conduction and wide working temperature.
Moreover, under electric field, along with Na^+ transport, some PC molecules are
confined into the sub-nanochannels due to the high interaction energy between
TPDBD-CNa and PC solvent, which can effectively prevent PC volatilization and
further promote Na^+ conduction (**Figure R13d-e**). It should be noted that the interaction
energy between TPDBD-CNa ($-11 \text{ Kcal mol}^{-1}$)/ Na^+ (-1 Kcal mol^{-1}) and PC were
calculated in TPDBD-CNa-NaTFSI/PC, and the stronger interaction energy between
PC and TPDBD-CNa framework promotes the adsorption of PC solvents at the sub-
nanometer zones (**Figure 13f**), and a very small amount of PC may form solvated
cations, and it was more difficult for the solvated PC to be removed from TPDBD-CNa-
NaTFSI and then undergo oxidation.

In this revision, we have supplemented this **Figure R12** and **Figure R13d-f** into the
Supplementary Information as **Supplementary Figure 25**, and also added
corresponding discussions the context and Supplementary Information to clarify this
issue in the manuscript (Please see Highlights in Page 9 and 10 of the revised
manuscript and Page 18 of Supplementary Information).

**Figure R9.** N₂ adsorption and desorption curves and pore size distributions of TPDBD-
CNa-NaTFSI and TPBD-NaTFSI.

Figure R10. Construction of negative electroactive zones. (a) Zeta potential values of various COFs. (b) Electrostatic potential mappings of TPBD, TPDBD, and TPDBD-COO⁻. (c) Optimized coordination structures between NaTFSI and TPBD, TPDBD-COO⁻. (d) Bond length between NaTFSI and TPBD, TPDBD and TPDBD-COO⁻.

Figure R11. Na⁺ transport mechanism of COF-NaTFSI. (a) Schematic of molecular simulation of TPDBD-CNa-NaTFSI (white, blue, cyan, pink, yellow, purple, and red balls denote hydrogen, nitrogen, carbon, fluorine, sulfur, sodium, and oxygen, respectively). (b) Na⁺ density mappings (Red regions indicate the highest probability

of Na⁺) of TPDBD-CNa-NaTFSI. (c) Na⁺ velocity at the edge and center of the TPDBD
 channel. O 1s (d), and F 1s (e) XPS spectra of TPDBD-CNa-QSSE/TPBD-QSSE
 obtained from Na||Na cells after 20 cycles and TPDBD-CNa-NaTFSI/TPBD-NaTFSI.

 **Figure R12.** (a) Thermodynamic equilibrium simulation of TPDBD-CNa-NaTFSI and
 PC. (b) Thermodynamic equilibrium simulation of TPBD-NaTFSI and PC.

 **Figure R13.** (a) FTIR spectra of the prepared TPDBD-CNa-QSSE, TPBD-QSSE, and
 PC solvent (cycled COF films in Na||Na cells after 20 cycles). (b) Na-O bond length of
 PC-Na and PC-NaTFSI. (c) TGA curves of the COF-based membranes and glass fiber
 with solvent. (d) Schematic of molecular simulation of TPDBD-CNa-NaTFSI/PC
 solvent (white, blue, cyan, red, and purple balls denote hydrogen, nitrogen, carbon,
 oxygen, and sodium, respectively. Brown area represents anion TFSI⁻). (e) MD
 simulation of PC distribution. (f) Interaction energy between TPDBD-CNa and PC, and
 TFSI⁻ in TPDBD-CNa-NaTFSI/PC solvent, and interaction energy between PC and
 Na⁺, and TFSI⁻ of (NaTFSI).

(4) Fig. S18 is hard to understand to distinguish the CIP and AGG, since the peaks are
so broad.

**Reply:** Many thanks for your important question. In order to distinguish the CIP and
AGG, we re-tested the Raman spectra many times. However, the specific peaks are
weak and it is really hard to divide due to the limited anion and cation aggregation
content. Even so, **Figure R14** shows an obvious blue shift in TPBD-NaTFSI and
TPDBD-CNa-NaTFSI compared with NaTFSI solution, indicating the increase of high
salt concentration consisting of cations/anions association and the formation of fast Na⁺
transport channels in COF-NaTFSI. The relevant description of Raman results has also
been revised in the manuscript (Please see Highlights in Page 7 of the revised
manuscript).

**Figure R14.** Raman spectra of TPDBD-CNa-NaTFSI, TPBD-NaTFSI, NaTFSI, and
solution.

(5) How about the performances of symmetrical cells and full cells, compared with the
previously reported values? Such as providing a table for compare.

**Reply:** Thanks for your valuable suggestion. We provide a table of the comparison of
performances of symmetrical cells and full cells with previously reported values
accordingly (**Table R2**). In our work, Na|TPDBD-CNa-QSSE|Na symmetrical cell
shows mid-to-upper Na stripping-plating performance at 25 °C without additional
pressure. Na₃V₂(PO₄)₃/C|QSSE|Na full cell shows long cycling stability over 1000
cycles with only 0.0048% capacity decay in each cycle, which is better than most of
the reported SSEs-based cells.

In this revision, we have supplemented **Table R2** into the Supplementary Information
as **Supplementary Table 5**, and also added corresponding discussions to manuscript to
clarify this issue (Please see Highlights in Page 16 of the revised manuscript and Page
40 of Supplementary Information).

**Table R2.** Comparison of symmetrical cells and full cells performance for the reported
 Li^+/Na^+ SSEs.

Materials	Li^+/Na^+	Solvent	Symmetrical cells		Cathodes	Full cells		Ref.
			Current densities (mA cm^{-2})	Time (h)		Current densities (A g^{-1})	Cycles/ Average capacity decay rate (%)	
$\text{LiPF}_6@$ ZIF-8	Li^+	45 wt.% EC/DMC/EMC	0.05	500	LiCoO_2	0.05	100/0.1190	1
UN-SLi	Li^+	33 wt.% EC/DEC	1	600	LiFePO_4	0.75	3000/0.0066	2
L@K/C	Li^+	NA	0.5	600	LiFePO_4	$\sim 0.0340/0.2$ C	300/0.052	3
$\text{LiPF}_6@$ PAF-1	Li^+	50 wt.% EC/DMC	NA	NA	LiFePO_4	0.68	1000/NA	4
CuBTC-PSS	Li^+	0.3 mg cm^{-2} (1M LiTFSI - PC)	1	600	LiFePO_4	$\sim 0.17/1$ C	500/NA	5
COF-NUST 7-8	Li^+	NA	0.05 (100 °C)	375	LiFePO_4	$\sim 0.0085/0.05$ C (100 °C)	60/0.1160	6
LGZ	Li^+	19.4 wt.% PC	0.1	500	LiFePO_4	0.17	500/ ~ 0	7
LiOOC-COF3	Li^+	10 μL LiPF_6 , EC/DEC v/v=1:1)	0.05	320	C_6O_6	0.05	500/0.02057	8
3D-UIO- 66/PAN/PEO/Li TFSI	Li^+	NA	0.3 (60 °C)	700	LiFePO_4	~ 0.34 (60 °C)	300/0.0467	9
DLC	Li^+	NA	0.3	450	LiFePO_4	$\sim 0.0017/0.1$ C (45 °C)	80/0.09615	10
NaFNFSI/ PEO	Na^+	NA	0.1 (80 °C)	200	$\text{NaCu}_{1/9}\text{Ni}_{2/9}\text{Fe}_{1/3}\text{Mn}_{1/3}\text{O}_2$	0.12 (80 °C)	150/0.2000	11
NaPTAB-SGPE	Na^+	66 wt.% PC	0.05 (60 °C)	100	$\text{Na}_3\text{V}_2(\text{PO}_4)_3$	0.06 (60 °C)	500/ ~ 0.0615	12
NaOOC-COF	Na^+	10 mL, 1.0 M NaPF_6/PC	0.05	700	BQ	0.2	600/0.0213	13
PLM@LE	Na^+	90 μL (NaClO_4 -PC with 5% FEC)	0.6	~ 300	$\text{Na}_{0.44}\text{MnO}_2$	0.1	160/0.0606	14
TPDBD-CNa- QSSE	Na^+	~ 9 wt.% solvent	0.01/0.05	1000/450	$\text{Na}_3\text{V}_2(\text{PO}_4)_3$	0.06/0.12	1000/0.0048 500/0.0097	This work

*Unless otherwise noted, tests were conducted at 25 °C.

References

- Sun, C. *et al.* ZIF-8-based quasi-solid-state electrolyte for lithium batteries. *ACS*
*Appl. Mater. Inter.* **11**, 46671-46677 (2019).
- Shi, W. *et al.* Electrolyte membranes with biomimetic lithium-ion channels.
*Nano Lett.* **20**, 5435-5442 (2020).
- Sun, W. *et al.* Ultrathin aramid/COF heterolayered membrane for solid-state Li-
metal batteries. *Nano Lett.* **20**, 8120-8126 (2020).
- Zou, J., Trewin, A., Ben, T. & Qiu, S. High uptake and fast transportation of
LiPF₆ in a porous aromatic framework for solid-state Li-ion batteries. *Angew.*
*Chem. Int. Ed. Engl.* **59**, 769-774 (2020).
- Chang, Z., Yang, H., Zhu, X., He, P. & Zhou, H. A stable quasi-solid electrolyte
improves the safe operation of highly efficient lithium-metal pouch cells in
harsh environments. *Nat. Commun.* **13**, 1510 (2022).
- Shan, Z. *et al.* Covalent organic framework-based electrolytes for fast Li⁺
conduction and high-temperature solid-state lithium-ion batteries. *Chem. Mater.*
**33**, 5058-5066 (2021).
- Jiang, G. *et al.* Glassy metal-organic-framework-based quasi-solid-state
electrolyte for high-performance lithium-metal batteries. *Adv. Funct. Mater.* **31**,
2104300 (2021).
- Zhao, G. *et al.* COF-based single Li⁺ solid electrolyte accelerates the ion
diffusion and restrains dendrite growth in quasi-solid-state organic batteries.
*Carbon Energy*, 1-13 (2022).
- Li, Z. *et al.* A 3D interconnected metal-organic framework-derived solid-state
electrolyte for dendrite-free lithium metal battery. *Energy Stor. Mater.* **47**, 262-
270 (2022).
- 10 Guo, D. *et al.* Foldable solid-state batteries enabled by electrolyte mediation in
covalent organic frameworks. *Adv. Mater.* **34**, 2201410 (2022).
- 11 Ma, Q. *et al.* A new Na[(FSO₂)(n-C₄F₉SO₂)N]-based polymer electrolyte for
solid-state sodium batteries. *J. Mater. Chem. A* **5**, 7738-7743 (2017).
- 12 Yang, L. *et al.* Novel sodium-poly(tartaric acid)borate-based single-ion
conducting polymer electrolyte for sodium-metal batteries. *ACS Appl. Energy*
*Mater.* **3**, 10053-10060 (2020).
- 13 Zhao, G. *et al.* COFs-based electrolyte accelerates the Na⁺ diffusion and
restrains dendrite growth in quasi-solid-state organic batteries. *Nano Energy* **92**,
106756 (2022).
- 14 Zhang, G. *et al.* Pancake-like MOF solid-state electrolytes with fast ion
migration for high-performance sodium battery. *Nano-Micro Lett.* **13**, 105
(2021).

(6) The cycling performances of the Na₃V₂(PO₄)₃/C|QSSEs|Na full battery (coin cells)
with a low cathode mass loading (1.3 mg cm⁻²) and a thick Na anode (300 μm) under
0.5 C is not enough for assessing its practical application. Commonly, the lower N/P
ratio or anode-free coin full cells is adopted to confirm the practical application of the
quasi-solid electrolyte in sodium batteries. Unfortunately, it cannot be found in this

*manuscript.*

**Reply:** Many thanks for your instructive comment. To further confirm the practical
application of the prepared QSSEs, we performed the cycling stability and rate
performance of Na₃V₂(PO₄)₃/C (NVP/C)|QSSEs|Na full cell with high cathode mass
loadings of active materials (2.6 mg cm⁻²) and low N/P ratio (~100 μm Na anode).
According to the editor's suggestion, the specific current (mA g⁻¹) values instead C-
rate values when electrochemical energy storage tests are discussed. 1 C charging and
discharging rate here was defined as 120 mA g⁻¹. As shown in **Figure R15**, the
NVP/C|TPDBD-CNa-QSSE|Na retains 63 mAh g⁻¹ after 100 cycles at 60 mA g⁻¹ with
a coulombic efficiency of 99.6%, and displays 99.7 mAh g⁻¹ after 68 cycles at 12 mA
g⁻¹ with a coulombic efficiency of 99.6%. When the current densities are 12, 24, 60,
and 120 mA g⁻¹, the specific capacities of NVP/C|TPDBD-CNa-QSSE|Na are 101.5,
106, 91.6 and 78.5 mAh g⁻¹, respectively. Even at a higher rate of 240 mA g⁻¹, the
battery still represents 54.5 mAh g⁻¹. When the current density recovers to 60 mA g⁻¹,
a specific capacity of 103.9 mAh g⁻¹ can be obtained, revealing reversible
characteristics. Those results verify that the NVP/C|TPDBD-CNa-QSSE|Na full cell
with high mass loading of cathode shows acceptable rate performance and cycle
stability.

Besides, we also assembled the anode-free battery by using a commercial carbon-
coated Al current collector as anode, TPDBD-CNa-QSSE as electrolyte, and NVP/C as
cathode. Unfortunately, when cycled at 12 mA g⁻¹, several assembled batteries all
exhibit similar capacity decay performance (**Figure R16**). The coulombic efficiency of
anode-free batteries is below 90%, which is the fatal factor for the practical application
of anode-free battery. The rapid capacity decline may be due to the unstable solid-
electrolyte interphase (SEI) layer by the large volume change of the deposited Na metal
and high dissolution of SEI components in the propylene carbonate (PC) electrolyte
(*Nature Energy* **7**, 511 (2022), *Angew. Chem. Int. Ed.* **61** e202200410, (2022), *ACS*
*Energy Lett.* **1**, 1173 (2016)). Actually, the reported anode-free sodium-ion batteries are
always assembled using hexafluoride phosphate (NaPF₆) and sodium tetrafluoroborate
(NaBF₄) in ether solvents to favor the high Na reversibility and low solubility of SEI.
Therefore, the electrochemical performance of the anode-free sodium-ion batteries is
closely related to the assembled components. In consideration of the promising
potential of anode-free Na⁺ batteries especially anode-free solid-state Na⁺ batteries,
great efforts will be paid for the deep studies of practical anode-free batteries in our
future works.

In this revision, we have supplemented this **Figure R15** into the Supplementary
Information as **Supplementary Figure 46**, and also added corresponding discussions
to both the context and Supplementary Information to clarify this issue (Please see
Highlights in Page 16 of the revised manuscript and Page 34 of Supporting Information).

Figure R15. (a) Photo of Na anode thickness. (b) Cycle stability at 60 mA g^{-1} , (c) cycle stability at 12 mA g^{-1} , and (d) rate performance of the NVP/C|TPDBD-CNa-QSSE|Na with high mass loading of active materials (2.6 mg cm^{-2}) in the cathode.

Figure R16. The cyclic stability of the anode-free battery (NVP/C|TPDBD-CNa-QSSE|Al@C) at 12 mA g^{-1} with 2.6 mg cm^{-2} mass loading.

(7) Moreover, pouch cells should be used in demonstrating the commercialization prospects of the quasi-solid electrolyte. Unfortunately, it also cannot be found in this manuscript. The evaluation is suggested to be provided.

Reply: Thank you so much for the valuable suggestion. We have investigated the electrochemical performance of pouch cells in the revised manuscript. In order to prepare the freestanding films with large areas, high flexibility, and mechanical strength, polytetrafluoroethylene (PTFE) was fibrillated to prepare large sheets of TPDBD-CNa

membranes according to the literature (*Nat Commun.* **13**, 1510 (2022), *Nano-Micro Lett.*
**13**,105 (2021), *Adv. Funct. Mater.* 2104300 (2021)). In detail, the mixture of TPDBD-
CNa, NaTFSI, and PTFE (Polytetrafluoroethylene dispersion) in a weight ratio of 8:1:1
in ethanol, and rolled into $\sim 90\ \mu\text{m}$ freestanding film and then cut into $\sim 5\times 5\ \text{cm}^2$ (**Figure**
**R17**). The bendable NVP/C|TPDBD-CNa-QSSE|Na pouch cell was assembled with a
$3\times 4\ \text{cm}^2$ rectangle cathode of 25.6 mg active materials and cycled at room temperature
(**Figure R18a-b**). The NVP/C|TPDBD-CNa-QSSE|Na pouch cell demonstrated
excellent cycling performance with $106.1\ \text{mAh g}^{-1}$ reserved after 65 cycles at a current
density of $12\ \text{mA g}^{-1}$) (**Figure R18c**). The LED device can function as usual under
different bending conditions, demonstrating the excellent flexibility and safety of the
bendable pouch cell. The use of TPDBD-CNa-QSSE would greatly promote the
development of safe and practical utilization of batteries.

In this revision, we have supplemented this **Figure R17**, **Figure R18b**, and **Figure**
**R18a, c** into the Supplementary Information and manuscript as **Supplementary Figure**
**48**, **Supplementary Figure 49**, and **Figure 8j-k**, respectively, and also added
corresponding discussions to both the context and Supplementary Information to clarify
this issue (Please see Highlights in Page 17 of the revised manuscript and Page 36 of
Supporting Information).

**Figure R17.** (a) Photos of the TPDBD-CNa membrane thickness at different positions.

(b) Photos of flexible freestanding solid TPDBD-CNa membrane.

**Figure R18.** (a) Schematic of the bendable NVP/C|TPDBD-CNa-QSSE|Na pouch cell.
 (b) Photo of a quasi-solid-state electrolyte pouch cell. (c) Cycling performance at 12
 572 mA g⁻¹ of NVP/C|TPDBD-CNa-QSSE|Na pouch cell (the insets are photos of a quasi-
 573 solid-state electrolyte pouch cell lighting LED).

(8) *The thickness of the membrane is too thick (400 μm), which adversely affects the*
 *energy density of the batteries. The decrease in thickness should be considered.*

**Reply:** Many thanks for your good suggestion, and we have performed the
 electrochemical performance of a full battery with a thin electrolyte membrane. **Figure**
 **R19a** shows the thickness of the electrolyte membrane was about 285 μm. As shown in
 **Figure R19b**, the NVP/C|TPDBD-CNa-QSSE|Na battery (active materials mass
 loading of 2.0 mg cm⁻²) exhibits 102.7 mAh g⁻¹ reserved after 150 cycles at a current
 density of 60 mA g⁻¹ with an average coulombic efficiency of ~99.7%. The battery
 assuredly exhibits excellent cycle life with reduced membrane thickness, which
 adversely affects the energy density of the batteries.

**Figure R19.** (a) Thicknesses of TPDBD-CNa-NaTFSI membrane. (b) Cycling
 performance and corresponding coulombic efficiency of NVP/C|TPDBD-CNa-
 QSSE|Na battery.

(9) How about the performances without adding any PC solvent, and in this case the
 intrinsically ionic conductivity can be evaluated?

**Reply:** Thank you very much for the constructive question. As shown in **Figure R20**,
 the Ti|TPDBD-CNa-NaTFSI|Ti without any PC solvent and addition pressure free was
 performed to investigate the intrinsically ionic conductivity from 25 to 100 °C. We find
 that the blocking titanium symmetric cell cannot work properly below 100 °C due to
 the weak contact between COF and titanium sheet. As the working temperature
 increased to 100 °C (**Figure R20c**), the intrinsic Na⁺ conductivity is about 4.49×10^{-6} S
 598 cm⁻¹. Therefore, the “Biomimetic Na⁺ Channel” of TPDBD-CNa can perform effective
 ion conduction even in solvent-free conditions.

In this revision, we have supplemented this **Figure R20** into the Supplementary
 Information as **Supplementary Figure 27**, and also added corresponding discussions
 to Supplementary Information to clarify this issue (Please see Highlights in Page 20 of
 Supporting Information).

**Figure R20.** (a) EIS of Ti|TPDBD-CNa-NaTFSI|Ti from 25 to 100 °C without solvent.
 (b) EIS of Ti|TPDBD-CNa-NaTFSI|Ti at 80 °C without solvent. (c) EIS of Ti|TPDBD-
 CNa-NaTFSI|Ti at 100 °C without solvent.

(10) For the specific surface area of TPDBD-CNa-NaTFSI and TPBD-NaTFSI are 38
and $28 \text{ m}^2 \text{ g}^{-1}$, respectively. And the weight loss of TPDBD-CNa and TPDBD are shown
as 44% and 46%, respectively. From the curves of Fig. S14, this could be due to
measurement error.

**Reply:** Thanks for your important comment. In order to eliminate the measurement
error, we have re-tested the N_2 adsorption and desorption of TPDBD-CNa-NaTFSI and
TPBD-NaTFSI (**Figure R21, Table R3**), and the TGA of TPDBD-CNa and TPDBD
for three times (**Figure R22**). The surface area of each COF-NaTFSI was calculated
using the BET model with relatively good fitting. The pore size distribution was
calculated based on the N_2 sorption isotherm by using the nonlocal density functional
theory (NLDFT, a carbon model containing slit pore) model in the Quantachrome
ASiQwin software. After filling NaTFSI, the BET surface area and total pore volume
of TPDBD-CNa-NaTFSI and TPBD-NaTFSI decrease to $16\text{-}21 \text{ m}^2 \text{ g}^{-1}$ and $0.07 \text{ cm}^3 \text{ g}^{-1}$
1 , $17\text{-}21 \text{ m}^2 \text{ g}^{-1}$ and $0.08 \text{ cm}^3 \text{ g}^{-1}$, respectively. TGA profiles of TPDBD showed the
decomposition temperature at $274 \text{ }^\circ\text{C}$ (43% weight loss), the first stage of weight loss
of TPDBD-CNa is mainly caused by the decomposition of $-\text{COO}^-$ at $264 \text{ }^\circ\text{C}$, and the
second stage begins at about $308 \text{ }^\circ\text{C}$ due to the collapse of framework (46% weight
loss). We have modified the results in our revised manuscript (Please see Highlights in
Page 7 of the revised manuscript).

**Figure R21.** (a-b) N_2 adsorption and desorption curves and pore size distributions of
TPDBD-CNa-NaTFSI tested three times. (c-d) N_2 adsorption and desorption curves and
pore size distributions of TPBD-NaTFSI tested three times.

**Table R3.** Comparison of BET surface area and total pore volume of TPDBD-CNa-
 NaTFSI and TPBD-NaTFSI.

	BET surface area (m ² g ⁻¹)	Total Pore Volume (cm ³ g ⁻¹)
TPDBD-CNa-NaTFSI-1	16	0.074
TPDBD-CNa-NaTFSI-2	19	0.072
TPDBD-CNa-NaTFSI-3	20	0.073
Average value (TPDBD-CNa-NaTFSI)	18	0.073
TPBD-NaTFSI-1	17	0.080
TPBD-NaTFSI-2	21	0.082
TPBD-NaTFSI-3	17	0.080
Average value (TPBD-NaTFSI)	18	0.081

**Figure R22.** TGA curves of TPDBD, TPDBD-CNa tested three times.

(11) For studying the chemical composition of the solid electrolyte interface (SEI), this
 manuscript only carried out the XPS experiments of the surface. But it is far from
 enough for the SEI analysis. For example, the XPS depth profiling analysis or the time-
 of-flight secondary ion mass spectrometry depth profiling analysis should be considered.

**Reply:** Thanks for your valuable comments. According to the reviewer's suggestions,
 we have further analyzed the cycled Na anode surface composition of Na|TPDBD-CNa-
 QSSE|Na battery to gain more insights into the SEI composition by conducting an in-
 depth Ar⁺ sputtering for 0, 40, 100, and 200 nm. As seen in the elemental distribution
 at different SEI depths (**Figure R23**), C and O elements are dominant on the surface,
 and the C element experiences a sharper decrease than O element upon sputtering. As
 the sputtering depth increases, the atomic percentage of Na element and Na/C ratio
 gradually increases, indicating a gradual approach to the Na anode surface.

Moreover, the C 1s, O 1s, F 1s, and Na 1s spectra were used to study the chemical
 composition in different depths of the solid electrolyte interface (SEI, **Figure R24**). As
 shown in the C 1s spectra, the fractional content of organic species such as C-C, C=O,
 RO-C=O, and C-F bond is detected and accompanied by inorganic species such as

Na₂CO₃ in the out SEI layer. As the sputtering depth increases, the peak intensity of
organic species tremendously decreases and the peak intensity of inorganic species
gradually increases. The variation trend is also detected in the O 1s spectra. When the
sputtering depth is 100 nm, the C–F peak is almost disappeared and the NaF signal
becomes stronger, indicating the dominant content of inorganic species such as NaF
and Na₂CO₃ in the inner layer of SEI. It should be noted that the C–F band can restrict
the Na⁺ transport (*Nano Energy* **102**, 107716 (2022)) and the NaF is beneficial for the
fast transport of Na⁺ (*Angew. Chem. Int. Ed.* **61**, e202200475 (2022)). Furthermore, the
O and F element content almost remains constant, indicating the homogeneous
distribution of SEI composition along with the various sputtering depth. In conclusion,
the SEI film of Na anode based on the TPDBD-CNa-QSSE is composed of fewer
organic carbonates and dense inorganic substances, which exhibits high mechanical
strength and is beneficial for the inhibition of dendrite growth (*ACS Energy Lett.* **7**,
2032–2042 (2022), *Research* **2022**, 9754612 (2022)).

In this revision, we have supplemented this **Figure R23** into the Supplementary
Information as **Supplementary Figure 36** and **Figure R24** into the manuscript as
**Figure 7b**, and also added corresponding discussions to manuscript to clarify this issue
(Please see Highlights in Page 13 and 14 of the revised manuscript).

**Figure R23.** Atomic ratio of the S, Na, F, O, C, and N elements in the SEI at various
sputtering depths of the cycled Na anode surface composition in Na|TPDBD-CNa-
QSSE|Na battery.

**Figure R24.** Comparison of the surface chemistry of cycled Na anodes. (a) C 1s, (b) O
 1s, (c) F 1s and (d) Na 1s depth profiles of Na|TPDBD-CNa-QSSE|Na after 20 cycles
 at 0.02 mA g^{-1} .

(12) For Table S2, the thickness of electrolyte should be also included for better
 comparison.

**Reply:** Thanks for your valuable suggestions. The thickness of electrolyte in **Table R4**
 is provided into the Supplementary Information as **Supplementary Table 4** (Please see
 Highlights in Page 39 of Supporting Information).

**Table R4.** Comparison of typical performance for the reported Li^+/Na^+ SSEs.

Materials	Li^+/Na^+	Solvent	Ion transference number	Electrochemical window (V)	Thickness (μm)	E_a (V)	σ (S cm^{-1})/ ($^\circ\text{C}$)	Ref.
ICOF-2/PC	Li^+	55 wt.% PC	0.8	4	NA	0.24	$3.05 \times 10^{-5}/(25)$	1
MIT-20-LiCl	Li^+	~70 wt.% PC	0.75	NA	NA	0.32	$1.3 \times 10^{-5}/(25)$	2
$\text{Li}^+/\text{Al-Td-MOF-1}$	Li^+	~50 wt.% PC	NA	NA	NA	0.1	$5.7 \times 10^{-5}/(25)$	3
$\text{LiPF}_6@ \text{ZIF-8}$	Li^+	45 wt.% EC/DMC/EMC	0.52	4.7	NA	0.458	$1.05 \times 10^{-5}/(25)$	4
UN-SLi	Li^+	33 wt.% EC/DEC	0.74	NA	NA	NA	$1.38 \times 10^{-5}/(25)$	5
L@K/C	Li^+	NA	NA	4.2	7.1	NA	$1.62 \times 10^{-4}/(30)$	6

LiPF ₆ @PAF-1	Li ⁺	50 wt.% EC/DMC	0.859	4.9	NA	0.15	4×10 ⁻⁴ /(25)	7
CuBTC-PSS	Li ⁺	0.3 mg cm ⁻² (1M LiTFSI-PC)	NA	5.4	38	0.096	5.75×10 ⁻⁴ /(25)	8
COF-NUST 7-8	Li ⁺	NA	0.11	4.2	NA	0.273	1.09×10 ⁻⁵ /(40)	9
LGZ	Li ⁺	19.4 wt.% PC	0.885	4.0	100	NA	1.61×10 ⁻⁴ /(30)	10
LiOOC-COF3	Li ⁺	NA	0.91	4.2	200	0.17	1.36×10 ⁻⁵ /(25)	11
3D-UIO-66/PAN/PEO/LiTFSI	Li ⁺	NA	0.52	NA	100	0.2	2.89×10 ⁻⁴ /(60)	12
DLC	Li ⁺	NA	0.85	4.5	32	NA	1.65×10 ⁻⁴ /(23)	13
MIT-20-Na	Na ⁺	Infiltrating with PC	NA	NA	NA	0.22	1.8×10 ⁻⁵ /(25)	14
NaNFSI/PEO	Na ⁺	NA	0.24	4.87	150	NA	3.36×10 ⁻⁴ /(80)	15
NaPTAB-SGPE	Na ⁺	66 wt.% PC	0.91(60)	5.2	50	0.127 (60)	9.4×10 ⁻⁵ /(25)	16
Ge-COF-Na	Na ⁺	PC	NA	NA	NA	0.25	3.4×10 ⁻⁵ /(100)	17
NaOOC-COF	Na ⁺	10.0 μL (1.0 M NaPF ₆ /PC)	0.9	4.2	200	0.24	2.68×10 ⁻⁴ /(25)	18
MIL-121/Na ⁺ SE	Na ⁺	50 wt.% (1 M NaClO ₄ /PC)	NA	NA	1000	0.36	1.2×10 ⁻⁴ /(30)	19
PLM@LE	Na ⁺	90 μL (NaClO ₄ -PC with 5% FEC)	0.33	4.8	120	0.112	6.6×10 ⁻⁵ /(25)	20
TPDBD-CNa-QSSE	Na ⁺	~9 wt.% solvent	0.90	5.32	400	0.204	1.30×10 ⁻⁴ /(25)	This work

*Unless otherwise noted, tests were conducted at 25 °C.

References

- Du, Y. *et al.* Ionic covalent organic frameworks with spiroborate linkage. *Angew. Chem. Int. Ed. Engl.* **55**, 1737-1741 (2016).
- Park, S. S., Tulchinsky, Y. & Dincă, M. Single-ion Li⁺, Na⁺, and Mg²⁺ solid electrolytes supported by a mesoporous anionic Cu-azolate metal-organic framework. *J. Am. Chem. Soc.* **139**, 13260-13263 (2017).
- Fischer, S. *et al.* A metal-organic framework with tetrahedral aluminate sites as a single-ion Li⁺ solid electrolyte. *Angew. Chem. Int. Ed.* **57**, 16683-16687 (2018).
- Sun, C. *et al.* ZIF-8-based quasi-solid-state electrolyte for lithium batteries. *ACS Appl. Mater. Inter.* **11**, 46671-46677 (2019).
- Shi, W. *et al.* Electrolyte membranes with biomimetic lithium-ion channels. *Nano Lett.* **20**, 5435-5442 (2020).
- Sun, W. *et al.* Ultrathin aramid/COF heterolayered membrane for solid-state Li-metal batteries. *Nano Lett.* **20**, 8120-8126 (2020).
- Zou, J., Trewin, A., Ben, T. & Qiu, S. High uptake and fast transportation of LiPF₆ in a porous aromatic framework for solid-state Li-ion batteries. *Angew. Chem. Int. Ed. Engl.* **59**, 769-774 (2020).

- Chang, Z., Yang, H., Zhu, X., He, P. & Zhou, H. A stable quasi-solid electrolyte
improves the safe operation of highly efficient lithium-metal pouch cells in
harsh environments. *Nat. Commun.* **13**, 1510 (2022).
- Shan, Z. *et al.* Covalent organic framework-based electrolytes for fast Li⁺
conduction and high-temperature solid-state lithium-ion batteries. *Chem. Mater.*
**33**, 5058-5066 (2021).
- Jiang, G. *et al.* Glassy metal–organic-framework-based quasi-solid-state
electrolyte for high-performance lithium-metal batteries. *Adv. Funct. Mater.* **31**,
2104300 (2021).
- Zhao, G. *et al.* COF-based single Li⁺ solid electrolyte accelerates the ion
diffusion and restrains dendrite growth in quasi-solid-state organic batteries.
*Carbon Energy*, 1-13 (2022).
- Li, Z. *et al.* A 3D interconnected metal-organic framework-derived solid-state
electrolyte for dendrite-free lithium metal battery. *Energy Stor. Mater.* **47**, 262-
270 (2022).
- 13 Guo, D. *et al.* Foldable solid-state batteries enabled by electrolyte mediation in
covalent organic frameworks. *Adv. Mater.* **34**, 2201410 (2022).
- 14 Park, S. S., Tulchinsky, Y. & Dincă, M. Single-ion Li⁺, Na⁺, and Mg²⁺ solid
electrolytes supported by a mesoporous anionic Cu–azolate metal–organic
framework. *J. Am. Chem. Soc.* **139**, 13260-13263 (2017).
- 15 Ma, Q. *et al.* A new Na[(FSO₂)(n-C₄F₉SO₂)N]-based polymer electrolyte for
solid-state sodium batteries. *J. Mater. Chem. A* **5**, 7738-7743 (2017).
- Yang, L. *et al.* Novel sodium–Poly(tartaric acid)borate-based single-ion
conducting polymer electrolyte for sodium–metal batteries. *ACS Appl. Energy*
*Mater.* **3**, 10053-10060 (2020).
- Ashraf, S. *et al.* Versatile platform of ion conducting 2D anionic germanate
covalent organic frameworks with potential for capturing toxic acidic gases.
*ACS Appl. Mater. Inter.* **12**, 40372-40380 (2020).
- Zhao, G. *et al.* COFs-based electrolyte accelerates the Na⁺ diffusion and
restrains dendrite growth in quasi-solid-state organic batteries. *Nano Energy* **92**,
106756 (2022).
- Zettl, R. *et al.* High Li⁺ and Na⁺ conductivity in new hybrid solid electrolytes
based on the porous MIL-121 metal organic framework. *Adv. Energy Mater.* **11**,
2003542 (2021).
- Zhang, G. *et al.* Pancake-like MOF solid-state electrolytes with fast ion
migration for high-performance sodium battery. *Nano-Micro Lett.* **13**, 105
(2021).

(13) Other minor issues should be taken seriously:

a) The abbreviation of Fourier transform infrared spectroscopy should be unify, such
as FT-IR in line 215 while FTIR in 162.

b) In line 192, the “a” in “-NaTFSI” should be in “Times New Roman” style.

c) The format of references should be uniform, such as title capitalization and journal
abbreviations.

**Reply:** Thanks for pointing out these mistakes. According to your comments, we have
carefully revised the manuscript. The abbreviation of Fourier transform infrared
spectroscopy is unified into FTIR (Please see Highlights in Page 10 and 20 of the
revised manuscript). The “a” in “-NaTFSI” is revised as “Times New Roman” style
(Please see Highlights in Page 8 of the revised manuscript). Moreover, we have
carefully checked the whole manuscript and unified the format of references in the
revised manuscript (Please see References of the revised manuscript and
Supplementary Information).

**Reviewer #3 (Remarks to the Author):**

*In this article, authors fabricated quasi-solid polymer electrolyte using TPBD-NaTFSI*
*and TPDBD-CNa-NaTFSI as the support substrate by absorbing PC and FEC solvent.*

*In this way, high ion conductivity and high oxidative potential is achieved. The strength*
*of this work is the biomimetic inspired synthesis of a new type of COF materials as the*
*host for Na ions conduction.*

*However, the weakness is also not insignificant. Specifically, because of the existence*
*of solvent, what determines the Na ions transport is the solution, unlike the case of ion*
*conduction via polymer chains in the polymer electrolytes. Authors repeatedly*
*emphasizes that the -COO- functional group anchored on the COF wall promotes the*
*decomposition of NaTFSI and boosts the Na ion transport. The calculation was*
*conducted based on this assumption. However, this assumption is questionable. Another*
*more possible reason to the enhanced ion conductivity is that the functionalized COF*
*has a stronger capability in retaining the solvent solution. The observed effect is not*
*limited to such COF. Instead, similar effect could also be realized from materials that*
*have a high specific surface area, porous structure and solvent-capturing functional*
*groups, for example, molecular sieves and OH- functionalized cellulose. The*
*conclusion will not be convincing unless authors can prove that the Na ions conduct in*
*the COF host much faster than in solution, and/or the effect of PC solvent to Na ion*
*conduction has been explicitly clarified.*

**Reply:** Thanks for your professional comments. According to the reviewer's instructive
suggestion, we have used more experimental and theoretical technologies to deeply
investigate the Na⁺ transport mechanism and PC effect. Hope the new result can give a
more convincing explanation.

**Firstly**, we investigate the Na⁺ conductivity of pure COF-NaTFSI without PC
addition and additional pressure at different temperatures. As shown in **Figure R25**,
due to the solid-solid contact between COF particles in the electrolyte membrane,
enormous grain boundary impedance can dominantly hinder the practical ionic transfer
across interfaces, resulting in a poor ionic conductivity of 4.49×10^{-6} S cm⁻¹ of pure
TPDBD. Even so, we can also find the advantage of biomimetic design in the Na⁺
transport because the pure COF of TPBD without Na⁺ sub-nanochannels design shows
almost no Na⁺ conductivity. The result is consistent with the verification in our
manuscript that the bionic design of Na⁺ transport channels in COFs can boost the rapid
dissociation of Na⁺ and TFSI⁻, and achieve selective rapid transport of Na⁺ along the
localized zones.

**Figure R25.** (a) EIS of Ti|TPDBD-CNa-NaTFSI|Ti from 25 to 100 °C without solvent.
 (b) EIS of Ti|TPDBD-CNa-NaTFSI|Ti at 80 °C without solvent. (c) EIS of Ti|TPDBD-
 CNa-NaTFSI|Ti at 100 °C without solvent. (d) EIS of Ti|TPBD-NaTFSI|Ti at 100 °C
 without solvent.

**Secondly**, we investigate the Na^+ conductivity of TPDBD with tiny PC solvent.
 Similar to other reported works, the PC solvent can effectively wet the grain boundary
 and improve the electrolyte/electrode interface compatibility, thus increasing the
 ultimate ion conduction properties with orders of magnitude improvement (*Nano-*
 *Micro Lett.* **13**, 105 (2021), *Nano Energy* **92**, 106756 (2022)). However, the big
 challenge in this field is how to effectively use solvent in the quasi-solid-state batteries,
 for instance furthest the performance with least solvent addition. As shown in **Table**
 **R5**, although relative high ion conductivity can be obtained in the reported works, the
 amount of PC addition was almost beyond 30 wt.%, which is very high and dominate
 the practical ionic transport. In our work, the result indicates that the ionic conductivity
 can reach as high as $1.30 \times 10^{-4} \text{ S cm}^{-1}$ at 25 °C when the amount of solvent addition is
 ~ 9 wt.%. However, without the biomimetic design, the ionic conductivity of TPBD is
 only $9.06 \times 10^{-5} \text{ S cm}^{-1}$. The result means that the biomimetic design of COFs shows
 great advantage in the efficient utilization of PC solvent.

**Finally**, in order to explicitly clarify the effect of PC solvent on Na ion conduction,
 molecular dynamic (MD) simulations of PC solvents on COFs surface, FTIR spectra of
 COFs with PC, DFT calculations of optimized structure, and TGA analysis of thermal
 stability were carried out. As shown in **Figure R26**, due to the strong wettability, tiny
 PC solvent can be rapidly adsorbed into COF membrane through thermodynamic

motion. MD simulation shows that the PC molecules are uniformly adsorbed and
infiltrated on the TPDBD-CNa skeletons while aggregated on TPBD surface, indicating
that the well-designed biomimetic COF is beneficial for the uniform wettability of PC
on the grain boundaries. The high-efficiency utilization of PC can improve the
electrode/electrolyte interface compatibility, and greatly reduce the interfacial
impedance between particles. In addition to the surficial wettability, along with Na⁺
transport under electric field, some PC molecules can be confined into the sub-
nanochannels due to the strong interaction between TPDBD-CNa and PC solvent,
which can effectively prevent PC volatilization and further promote the Na⁺ transport
in the biomimetic sub-nanochannels (**Figure R27a-b**). It should be noted that the
interaction energy between TPDBD-CNa (-11 Kcal mol⁻¹)/ Na⁺(-1 Kcal mol⁻¹) and PC
were calculated in TPDBD-CNa-NaTFSI/PC, and the stronger interaction energy
between PC and TPDBD-CNa framework promotes the adsorption of PC solvents at
the sub-nanometer zones (**Figure R27c**), and a very small amount of PC may form
solvated cations, and it was more difficult for the solvated PC to be removed from
TPDBD-CNa-NaTFSI and then undergo oxidation. As shown in **Figure R27d**,
TPDBD-CNa-QSSE behaves stronger propylene carbonate (PC)-Na⁺ interactions (720
and 1403 cm⁻¹) and concentrated TFSI⁻ (787 cm⁻¹) than TPBD-QSSE, demonstrating
stronger electrolyte aggregation effect (*Nat. Commun.* **13**, 1510 (2022)). DFT
calculations reveal that the Na-O bond length of NaTFSI becomes significantly longer,
and PC displays strong interaction with Na⁺, indicating PC can effectively dissociate
Na⁺ and TFSI⁻ (**Figure R27e**). TGA spectra of COF-based membranes with solvent
shows higher thermal stability than pure solvent and glass fiber/solvent (**Figure R27f**),
further verifying the strong confinement of PC in the COF interior. The high stability
of QSSEs is advantageous for the elimination of grain boundary impedance, thus
effectively boosting the Na⁺ conductivity and wide working temperature.

**Table R5.** Comparison of typical performance for the reported Li⁺/Na⁺ SSEs.

Materials	Li ⁺ /Na ⁺	Solvent	Ion transference number	Electrochemical window (V)	Thickness (μm)	Ea (V)	σ (S cm ⁻¹)/(°C)	Ref.
ICOF-2/PC	Li ⁺	55 wt.% PC	0.8	4	NA	0.24	3.05×10 ⁻⁵ /(25)	1
MIT-20-LiCl	Li ⁺	~70 wt.% PC	0.75	NA	NA	0.32	1.3×10 ⁻⁵ /(25)	2
Li ⁺ /Al-Td-MOF-1	Li ⁺	~50 wt.% PC	NA	NA	NA	0.1	5.7×10 ⁻⁵ /(25)	3
LiPF ₆ @ ZIF-8	Li ⁺	45 wt.% EC/DMC/EMC	0.52	4.7	NA	0.458	1.05×10 ⁻⁵ /(25)	4
UN-SLi	Li ⁺	33 wt.% EC/DEC	0.74	NA	NA	NA	1.38×10 ⁻⁵ /(25)	5
LiPF ₆ @PAF-1	Li ⁺	50 wt.% EC/DMC	0.859	4.9	NA	0.15	4×10 ⁻⁴ /(25)	6
LGZ	Li ⁺	19.4 wt.% PC	0.885	4.0	100	NA	1.61×10 ⁻⁴ /(30)	7
NaPTAB-SGPE	Na ⁺	66 wt.% PC	0.91(60)	5.2	50	0.127	9.4×10 ⁻⁵ /(25)	8
NaOOC-COF	Na ⁺	10.0 μL (1.0 M NaPF ₆ /PC)	0.9	4.2	200	0.24	2.68×10 ⁻⁴ /(25)	9
MIL-121/Na ⁺ SE	Na ⁺	50 wt.% (1 M NaClO ₄ /PC)	NA	NA	1000	0.36	1.2×10 ⁻⁴ /(30)	10

PLM@LE	Na ⁺	90 μL (NaClO ₄ -PC with 5% FEC)	0.33	4.8	120	0.112	6.6×10 ⁻⁵ /(25)	11
TPDBD-CNa-QSSE	Na ⁺	~9 wt.% solvent	0.90	5.32	400	0.204	1.32 ×10 ⁻⁴ /(25)	This work

References

- Du, Y. *et al.* Ionic covalent organic frameworks with spiroborate linkage.
*Angew. Chem. Int. Ed. Engl.* **55**, 1737-1741 (2016).
- Park, S. S., Tulchinsky, Y. & Dincă, M. Single-ion Li⁺, Na⁺, and Mg²⁺ solid
electrolytes supported by a mesoporous anionic Cu–Azolate metal–organic
framework. *J. Am. Chem. Soc.* **139**, 13260-13263 (2017).
- Fischer, S. *et al.* A metal–organic framework with tetrahedral aluminate sites as
a single-ion Li⁺ solid electrolyte. *Angew. Chem. Int. Ed.* **57**, 16683-16687,
(2018).
- Sun, C. *et al.* ZIF-8-based quasi-solid-state electrolyte for lithium batteries. *ACS*
*Appl. Mater. Inter.* **11**, 46671-46677 (2019).
- Shi, W. *et al.* Electrolyte membranes with biomimetic lithium-ion channels.
*Nano Lett.* **20**, 5435-5442 (2020).
- Zou, J., Trewin, A., Ben, T. & Qiu, S. High uptake and fast transportation of
LiPF₆ in a porous aromatic framework for solid-state Li-Ion batteries. *Angew.*
*Chem. Int. Ed. Engl.* **59**, 769-774 (2020).
- Jiang, G. *et al.* Glassy metal–organic-framework-based quasi-solid-state
electrolyte for high-performance lithium-metal batteries. *Adv. Funct. Mater.* **31**,
2104300 (2021).
- Yang, L. *et al.* Novel sodium–poly(tartaric acid)borate-based single-ion
conducting polymer electrolyte for sodium–metal batteries. *ACS Appl. Energy*
*Mater.* **3**, 10053-10060 (2020).
- Zhao, G. *et al.* COFs-based electrolyte accelerates the Na⁺ diffusion and
restrains dendrite growth in quasi-solid-state organic batteries. *Nano Energy* **92**,
106756 (2022).
- Zettl, R. *et al.* High Li⁺ and Na⁺ conductivity in new hybrid solid electrolytes
based on the porous MIL-121 metal organic framework. *Adv. Energy Mater.* **11**,
2003542 (2021).
- Zhang, G. *et al.* Pancake-like MOF solid-state electrolytes with fast ion
migration for high-performance sodium battery. *Nano-Micro Lett.* **13**, 105
(2021).

**Figure R26.** (a) Photo of SSE with ~9 wt.% solvents (No-flowing liquid on the surface).

(b) Thermodynamic equilibrium simulation of TPDBD-CNa-NaTFSI and PC. (c)

Thermodynamic equilibrium simulation of TPBD-NaTFSI and PC.

**Figure R27.** (a) Schematic of molecular simulations of TPDBD-CNa-NaTFSI/PC
 solvent (white, blue, cyan, red, and purple balls denote hydrogen, nitrogen, carbon,
 oxygen, and sodium, respectively. Brown area represents anion TFSI⁻).

(b) MD simulation of PC distribution. (c) Interaction energy between TPDBD-CNa and PC, and
 TFSI⁻ in TPDBD-CNa-NaTFSI/PC solvent, and interaction energy between PC and
 Na⁺, and TFSI⁻ of (NaTFSI).

(d) FTIR spectra of the prepared TPDBD-CNa-QSSE, TPBD-QSSE, and PC solvent (cycled COF
 films in Na||Na cells after 20 cycles).

(e) Na-O bond length of PC-Na and PC-NaTFSI. (f) TGA curves of the COF-based
 membranes and glass fiber with solvent.

Besides, to further verify the excellent matching between the tiny addition of PC and
 biomimetic COF skeleton, additional experiments of QSSEs at low temperatures were
 carried out. As shown in **Figure R28a**, as the temperature decreased from 0 to -20 °C,
 Na|TPDBD-CNa-QSSE|Na represents low overpotential and stable voltage profiles

under 0.01 mA cm^{-2} . When the temperature is further decreased to the severe $-25 \text{ }^\circ\text{C}$,
 irregular voltage fluctuations emerge, which can be ascribed to the unstable Na
 plating/stripping and formation of dead Na (*Nano Lett.* **19**, 8664 (2019), *ACS Energy*
 *Lett.* **5**, 2411 (2020)). However, due to the absence of biomimetic sub-nanochannels,
 the voltage irregularity of Na|TPBD-QSSE|Na symmetrical cell appears at relative
 higher temperature of $-10 \text{ }^\circ\text{C}$. Moreover, **Figure R28b** exhibits relatively stable voltage
 curves for over 550 h without obvious overpotential fluctuation at $0 \text{ }^\circ\text{C}$, further
 indicating that the bioinspired design of ionic channels is beneficial for the uniform
 deposition/stripping of Na^+ . In addition, the NVP/C|TPDBD-CNa-QSSE|Na cell (active
 material mass loading of 2.0 mg cm^{-2}) at $0 \text{ }^\circ\text{C}$ shows 101.2 mAh g^{-1} reserved after 75
 cycles at 12 mA g^{-1} with a coulombic efficiency of 99.4% (**Figure R28c**), confirming
 excellent cycling stability of QSSE based full cell at decreased temperatures. Therefore,
 it is concluded that TPDBD-CNa-based QSSE can effectively operate at lower
 temperatures and shows admirable adaptability to temperature.

**Figure R28.** (a) Temperature-dependent galvanostatic cycling of Na stripping-plating
 test of Na|QSSEs|Na at 0, -5, -10, -15, -20, and -25 °C with a current density of 0.01
 920 mA cm^{-2} . (b) Long cycle performance of Na|TPDBD-CNa-QSSE|Na at $0 \text{ }^\circ\text{C}$ with a
 921 current density of 0.01 mA cm^{-2} . (c) Cycle performance of NVP|TPDBD-CNa-
 922 QSSE|Na at 12 mA g^{-1} at $0 \text{ }^\circ\text{C}$.

**Furthermore**, to assess the practical application of biomimetic design concept, we
 also carried out the full battery test under the high mass loading of NVP/C and thin
 QSSEs membrane with tiny PC addition in both coin and pouch batteries. According to
 the editor's suggestion, the specific current (mA g^{-1}) values instead C-rate values when
 electrochemical energy storage tests are discussed. 1 C charging and discharging rate
 here was defined as 120 mA g^{-1} . As shown in **Figure R29**, the NVP/C|TPDBD-CNa-

QSSE|Na retains 63 mAh g⁻¹ after 100 cycles at 60 mA g⁻¹ with a coulombic efficiency
 of 99.6%, and displays 99.7 mAh g⁻¹ after 68 cycles at 12 mA g⁻¹ with a coulombic
 efficiency of 99.6%. When the current densities are 12, 24, 60, and 120 mA g⁻¹, the
 specific capacities of NVP/C|TPDBD-CNa-QSSE|Na are 101.5, 106, 91.6 and 78.5
 mAh g⁻¹, respectively. Even at a higher rate of 240 mA g⁻¹, the battery still represents
 54.5 mAh g⁻¹. When the current density recovers to 60 mA g⁻¹, a specific capacity of
 103.9 mAh g⁻¹ can be obtained, revealing reversible characteristics. Those results verify
 that the NVP/C|TPDBD-CNa-QSSE|Na full cell with high mass loading of cathode
 shows acceptable rate performance and cycle stability. In order to prepare the
 freestanding films with large areas, high flexibility, and mechanical strength,
 polytetrafluoroethylene (PTFE) was fibrillated to prepare large sheets of TPDBD-CNa
 membranes according to the literature (*Nat Commun.* **13**, 1510 (2022), *Nano-Micro Lett.*
 **13**,105 (2021), *Adv. Funct. Mater.* 2104300 (2021)). In detail, the mixture of TPDBD-
 CNa, NaTFSI, and PTFE (Polytetrafluoroethylene dispersion) in a weight ratio of 8:1:1
 in ethanol, and rolled into ~90 μm freestanding film and then cut into ~5×5 cm² (**Figure**
 **R30**). The bendable NVP@C|TPDBD-CNa-QSSE|Na pouch cell was assembled with
 a 3×4 cm² rectangle cathode of 25.6 mg active materials and cycled at room temperature
 (**Figure R31a-b**). The NVP@C|TPDBD-CNa-QSSE|Na pouch cell demonstrated
 excellent cycling performance with 106.1 mAh g⁻¹ reserved after 65 cycles at a current
 density of 12 mA g⁻¹) (**Figure R31c**). The LED device can function as usual under
 different bending conditions, demonstrating the excellent flexibility and safety of the
 bendable pouch cell. The use of TPDBD-CNa QSSE would greatly promote the
 development of safe and practical utilization of batteries.

**Figure R29.** (a) Photo of Na anode thickness. (b) Cycle stability at 60 mA g⁻¹, (c) cycle
 stability at 12 mA g⁻¹, and (d) rate performance of the NVP/C|TPDBD-CNa-QSSE|Na
 with high mass loading of active materials (2.6 mg cm⁻²) in the cathode.

**Figure R30.** (a) Photos of the TPDBD-CNa membrane thickness at different positions.

(b) Photos of flexible freestanding solid TPDBD-CNa membrane.

**Figure R31.** (a) Schematic of the bendable NVP/C|TPDBD-CNa-QSSE|Na pouch cell.

(b) Photo of a quasi-solid-state electrolyte pouch cell. (c) Cycling performance at 12

968 mA g⁻¹ of NVP/C|TPDBD-CNa-QSSE|Na pouch cell (the inset is photos of a quasi-

969 solid-state electrolyte pouch cell lighting LED).

**In conclusion**, the well-designed biomimetic Na⁺ nanochannels are of great
importance for the final performance of QSSEs, especially at extremely low
temperatures. Moreover, we believe that how to effectively use solvent in the quasi-
solid state batteries may attract increasing attention in the future, for instance, furthest
the performance with least solvent addition. Great efforts will be paid for the deep
studies of quasi-solid-state sodium batteries in our future works, particularly the
atomic-level matching between solvent and skeletons.

**Other minor issues:**

*1) it is not meaningful to test the thermal stability of pure TPBD-NaTFSI and TPDBD-*
*CNa-NaTFSI; instead, it is more relevant for battery practice to test the thermal*
*stability of their mixture with solvent.*

**Reply:** We are thankful for the reviewer's valuable suggestion. According to the
reviewer's comment, we have investigated the thermal stability of TPBD-NaTFSI and
TPDBD-CNa-NaTFSI with solvent. Due to the sub-nanometer confinement effects
promoted by the sub -nanochannels of the COF, the ~9 wt.% solvent (PC with 5% FEC)
tended to be attached to the interior of COFs and decomposed more slowly than pure
solvent and glass fiber/solvent (**Figure R31**), making it harder to oxidize and
decompose the quasi-solid-state electrolyte, and further verifying the excellent
interfacial contact and compatibility of PC confined in the QSSEs, thus effectively
boosting the Na⁺ conduction and wide working temperature.

In this revision, we have supplemented this **Figure R32** into the Supplementary
Information as **Supplementary Figure 26**, and also added corresponding discussions
to manuscript to clarify this issue (Please see Highlights in Page 10 of the revised
manuscript).

**Figure R32.** TGA curves of the COF-based membranes and glass fiber with solvent.

2) the LSV test shows a wide electrochemical stability window of 5.32 V. This is good
for the electrolyte membrane itself but it does not mean the full-cell battery can deliver
such high voltage. The actual output voltage as well as power performance of a Na-ion
battery based on quasi-solid polymer electrolytes is more significantly affected by the
interface property. So it could be misleading to claim a high-performance full cell based
on high-performance of the electrolyte membrane.

**Reply:** Thanks for your valuable comments. We are fully agreed with the reviewer's
suggestion that a high-performance full cell is inseparable from the good interface
between electrodes and a quasi-solid-state electrolyte (QSSE). In our work, the high-
efficiency utilization of tiny PC solvent can effectively improve the compatibility of
electrode/electrolyte interface, and greatly reduce the interfacial impedance (**Figure**
**R26**). Besides, some PC molecules can be confined into the sub-nanochannels, which
can further prevent PC volatilization and enhance the stability of QSSE (**Figure R27**).
Based on the systematic improvement, the full battery shows high performance even
with high mass loading and under low-temperatures (<0 °C, **Figure R28-31**).
According to the reviewer's suggestion, we have modified the corresponding
description of the battery application section in our revised manuscript.

REVIEWERS' COMMENTS

Reviewer #1 (Remarks to the Author):

The authors improved the manuscript over their original version. I recommend publishing the manuscript.

Reviewer #2 (Remarks to the Author):

I am satisfied with this round revision since all my concerns have been well addressed. Therefore, it can be accepted as it is.

Reviewer #3 (Remarks to the Author):

The authors have very carefully addressed all comments by the three reviewers and provided new data to further strengthen the conclusion. Given the significant improvement in data completeness and reasonable justification in the response letter, I would like to recommend acceptance of this paper for Nature Commun.